# SpaCeFormer: Fast Proposal-Free
# Open-Vocabulary 3D Instance Segmentation

**Chris Choy** [1]  **Junha Lee** [2]  **Chunghyun Park** [2]  **Minsu Cho** [2]  **Jan Kautz** [1]

## Abstract

Open-vocabulary 3D instance segmentation is a core capability for robotics and AR/VR, but prior methods trade one bottleneck for another: multi-stage 2D+3D pipelines aggregate foundation-model outputs at hundreds of seconds per scene, while pseudo-labeled end-to-end approaches rely on fragmented masks and external region proposals. We present SpaCeFormer, a proposal-free space-curve transformer that runs in 0.12–0.30 seconds per scene across standard benchmarks, 2–3 orders of magnitude faster than multi-stage 2D+3D pipelines. We pair it with SpaCeFormer-3M, the largest open-vocabulary 3D instance segmentation dataset (3.0M multi-view-consistent captions over 604K instances from 7.4K scenes) built through multi-view mask clustering and multi-view VLM captioning; it reaches **21× higher mask recall** than prior single-view pipelines (54.3% vs 2.5% at IoU>0.5). SpaCeFormer combines spatial window attention with Morton-curve serialization for spatially coherent features, and uses a RoPE-enhanced decoder to predict instance masks directly from learned queries without external proposals. On ScanNet200 we achieve 11.1 zero-shot mAP, a 2.8× improvement over the prior best proposal-free method; on ScanNet++ and Replica, we reach 22.9 and 24.1 mAP, surpassing all prior methods including those using multi-view 2D inputs.

## 1. Introduction

Understanding 3D scenes with open-vocabulary descriptions is fundamental for autonomous driving (Zhang et al., 2024), augmented reality (Guo et al., 2024), and embodied AI (Rashid et al., 2023; Driess et al., 2023). Unlike

[1]NVIDIA [2]POSTECH. Correspondence to: Chris Choy <cchoy@nvidia.com>.

*Proceedings of the 43rd International Conference on Machine Learning*, Seoul, South Korea. PMLR 306, 2026. Copyright 2026 by the author(s).

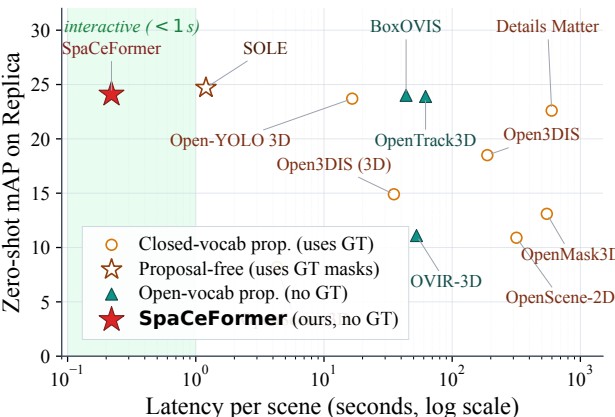

*Figure 1.* **Accuracy vs. latency on Replica (zero-shot).** Filled markers: open-vocab, no GT 3D supervision. Hollow: uses GT 3D supervision. SpaCeFormer (red star) reaches **24.1 mAP at 0.22 s** – Pareto-optimal without GT 3D, 2–3 orders faster than multi-stage 2D+3D pipelines, and the only method in the interactive band (<1 s, shaded). Log-scale latency; full numbers in Tab. 3.

closed-set 3D segmentation (Qi et al., 2017; Choy et al., 2019; Schult et al., 2023) that assumes a fixed label space, open-vocabulary 3D segmentation aims to identify and segment arbitrary objects described in natural language (Peng et al., 2023). This capability is crucial in real-world environments where the long-tail of object types far exceeds any predefined taxonomy (Liu et al., 2019).

The lack of large-scale 3D-text paired data (Yang et al., 2024a) has driven two parallel responses: (1) multi-stage pipelines that aggregate 2D foundation model proposals (Kirillov et al., 2023; Radford et al., 2021) into 3D (Takmaz et al., 2023; Nguyen et al., 2024; Yin et al., 2024), and (2) pseudo-labeled 3D-text datasets for end-to-end training (Luo et al., 2024; 2023; Jia et al., 2024; Yang et al., 2024b; Lee et al., 2025a). Multi-stage pipelines suffer compounding errors, dataset-specific thresholds, and hundreds of seconds of latency per scene (Fig. 1); pseudo-labeling inherits fragmented 3D instances from per-view lifting and inconsistent descriptions from single-view captioning. Compounding both, open-vocabulary instance segmentation demands spatially coherent features for sharp boundaries, yet 3D backbones like Point Transformer (Wu et al., 2024) and OctFormer (Wang, 2023) serialize via space-filling curves that scatter spatially adjacent points across attention win-

dows, sacrificing the local coherence that instance segmentation needs to recover sharp object boundaries.

To address these limitations in pseudo-labeled dataset construction and 3D backbone design, we introduce SpaCeFormer-3M, a large-scale open-vocabulary 3D instance segmentation dataset constructed via multi-view mask clustering and captioning, and SpaCeFormer (**Spa**ce-**C**urve Trans**former**), an end-to-end network architecture tailored for open-vocabulary 3D instance segmentation. Our key insight is that multi-view consistency, both in mask aggregation and captioning, combined with spatially coherent 3D features enables effective proposal-free instance segmentation without multi-stage heuristics.

On the data side, we address two key limitations of RegionPLC (Yang et al., 2024b) and Mosaic3D (Lee et al., 2025a): fragmented masks and inconsistent captions caused by single-view processing. We aggregate partial 2D masks into complete, geometry-consistent 3D instances via multi-view mask clustering (Yan et al., 2024), and employ multi-view prompting to produce view-consistent descriptions. The resulting dataset, SpaCeFormer-3M (3.0M multi-view-consistent captions over 604K instance masks across 7.4K scenes), has significantly higher mask completeness and caption consistency than prior pseudo-label corpora: **54.3% mask recall at IoU>0.5** on ScanNet versus 2.5% for single-view pipelines like Mosaic3D, a $21\times$ improvement (Sec. 3). Geometry-consistent training data is essential for our proposal-free architecture: the decoder must learn to predict complete instance masks from spatially coherent features, which requires training on complete, non-fragmented mask annotations like those in SpaCeFormer-3M.

On the modeling side, SpaCeFormer builds on Point Transformer v3 (Wu et al., 2024) with a new attention design: *space-curve attention*, which combines spatial window attention with Morton curve serialization. Each component exists in prior work, but their combination is novel and specifically motivated by our proposal-free setting. Spatial windows preserve coherent local neighborhoods that matter for instance boundary prediction (achieving 28.6% lower intra-window spatial distance than Morton-only attention), while Morton curves provide structured diversity at coarser scales. Together with 3D RoPE (Su et al., 2024) in both backbone and decoder, these features support proposal-free mask prediction via learned queries (Carion et al., 2020; Cheng et al., 2022) without multi-stage heuristics.

We evaluate SpaCeFormer on three benchmarks for open-vocabulary 3D instance segmentation. On Replica, SpaCeFormer reaches **24.1 zero-shot mAP at 0.22 s per scene**, the only open-vocabulary method that operates at interactive rates; in accuracy it is matched only by SOLE (Lee et al., 2025b) (24.7 mAP), which requires ScanNet200 ground-truth mask supervision and is $\sim 9\times$ slower (Fig. 1, Tab. 3).

On ScanNet200 (Tab. 5), under the matched regime of *proposal-free, 3D-only input, no GT 3D annotations*, SpaCeFormer attains 11.1 mAP at 0.12 s per scene, $2.8\times$ over the next-best method in this regime (Mosaic3D+Decoder, 3.9 mAP). Methods that score higher on ScanNet200 rely on either GT-trained proposals (*e.g.* Mask3D-based pipelines at up to 24.7 mAP, 16–554 s) or multi-view 2D streams with YOLO/SAM proposals (up to 26.0 mAP, $\sim 356$ s). On ScanNet++, we achieve 22.9 mAP, surpassing the best prior method OpenTrack3D (20.6 mAP) which relies on multi-view 2D inputs and requires $\sim 320$ seconds per scene—over $2{,}000\times$ slower than ours (Tab. 4).

Our contributions are four-fold:

- SpaCeFormer-3M, the largest open-vocabulary 3D instance segmentation dataset (604K masks, 3.0M multi-view-consistent captions across 7.4K scenes), with $21\times$ higher mask recall (54.3% vs 2.5% at IoU>0.5) than prior single-view pseudo-label pipelines.
- SpaCeFormer, a space-curve attention backbone combining sliding 3D window attention with Morton curve serialization for spatially coherent features.
- A RoPE-enhanced proposal-free decoder that predicts instance masks directly from learned queries without external heuristics.
- State-of-the-art results on ScanNet200 and ScanNet++ with strong zero-shot transfer across datasets.

## 2. Related Work

**3D Instance Segmentation.** Closed-vocab methods (Mask3D (Schult et al., 2023), ISBNet (Ngo et al., 2023)) train proposal networks on fixed taxonomies. Open-vocab methods fuse 3D proposals with CLIP (Radford et al., 2021) features via multi-stage pipelines: OpenMask3D (Takmaz et al., 2023) aggregates per-proposal multi-view CLIP features; Open3DIS (Nguyen et al., 2024) merges 2D and 3D proposals hierarchically; SAI3D (Yin et al., 2024) lifts 2D masks to 3D; Open-YOLO 3D (Boudjoghra et al., 2025) aligns proposals with open-vocab 2D detectors. These depend on heavy 2D components (SAM (Kirillov et al., 2023), Grounded-SAM (Ren et al., 2024)) and repeated multi-view projections, accumulating cost and compounding errors. SOLE (Lee et al., 2025b) avoids explicit proposals but still requires GT 3D mask supervision. We instead train a proposal-free decoder entirely on pseudo-labels.

**Point Cloud Architectures.** Sparse convolutions (Choy et al., 2019; Thomas et al., 2019) and point transformers (Wu et al., 2022; 2024; Park et al., 2022) dominate 3D understanding. PTv3 (Wu et al., 2024) serializes points by Morton codes for scalable attention but scatters spatially adjacent points across attention windows, sacrificing the local coherence instance segmentation needs for sharp boundaries.

Our space-curve attention combines spatial window attention (locality) with Morton serialization (diversity). We further adopt 3D RoPE (Su et al., 2024), extending the 1D formulation to encode 3D spatial relationships in attention.

**Scalable 3D Annotation.** Open-vocab 3D training needs mask–text pairs, prohibitive to annotate manually. Cap3D (Luo et al., 2023) aggregates multi-view captions for assets; SceneVerse (Jia et al., 2024) scales vision–language data for indoor scenes; Mosaic3D (Lee et al., 2025a) aligns millions of 3D masks with text. But 2D-to-3D lifting fragments instances and single-view captioning yields inconsistent descriptions. Multi-view mask clustering (Yan et al., 2024) addresses fragmentation and ranked view selection (Luo et al., 2024) improves consistency; we combine both in SpaCeFormer-3M.

## 3. Dataset Generation

Manual mask–language annotation at the scale needed for open-vocabulary 3D training is prohibitive, so recent pipelines (Yang et al., 2024b; Jiang et al., 2024; Lee et al., 2025a) use 2D foundation models, but suffer from (1) fragmented 3D masks from naive 2D-to-3D lifting and (2) inconsistent captions from single-view prompting. We address both via multi-view mask clustering for geometry-consistent instances and structured multi-view captioning for view-consistent descriptions (Figure 2).

### 3.1. Data Sources and Preprocessing

We aggregate four large-scale datasets spanning diverse capture conditions: ScanNet (Dai et al., 2017), ScanNet++ (Yeshwanth et al., 2023), Matterport3D (Chang et al., 2018), and ARKitScenes (Baruch et al., 2021). This combination covers professional high-quality scans (ScanNet++) and casual mobile captures (ARKitScenes), as well as diverse indoor environments (ScanNet, Matterport3D), ensuring the model generalizes across reconstruction qualities and scene types. All data are standardized into a unified format with calibrated intrinsics and poses (see Appendix A.6).

### 3.2. Annotation Pipeline

**Step 1: Efficient View Sampling.** Video datasets contain redundant frames. We greedily select frames whose geometric overlap with the most recently selected frame falls below a threshold, measured as the fraction of points in the current frame with a nearest neighbor in the reference frame within a distance bound (full definition: Appendix A.6). This maximizes coverage while removing redundancy.

**Step 2: Training-free 3D Mask Aggregation.** Per-view 2D-to-3D lifting fragments masks because each view captures only partial surfaces. We detect 2D masks with SAM2 (Ravi et al., 2024), lift them to 3D, and apply

*Table 1.* **Dataset statistics for SpaCeFormer-3M.** Our training corpus comprises 7,361 scenes with over 604K 3D instance masks and 3.0M captions aggregated from diverse indoor RGB-D datasets (ScanNet, ScanNet++, ARKitScenes, Matterport3D). Each mask is annotated with 5 captions from different viewpoints using our multi-view-aware captioning pipeline. "Avg masks/scene" indicates the average number of instance masks per scene, reflecting varying scene complexity across datasets.

| Dataset | Scenes | Masks | Captions | Words | Avg masks/scene |
|---|---|---|---|---|---|
| ScanNet | 1,201 | 79,320 | 396,600 | 6,105,677 | 66.04 |
| ScanNet++ | 223 | 27,296 | 136,480 | 2,132,620 | 122.40 |
| ARKitScenes | 4,497 | 446,409 | 2,232,045 | 34,374,338 | 99.27 |
| Matterport3D | 1,440 | 51,102 | 255,510 | 3,958,785 | 35.49 |
| **Overall** | **7,361** | **604,127** | **3,020,635** | **46,571,420** | **82.07** |

*Table 2.* **3D mask quality vs. ground truth (312 ScanNet train scenes).** We compare auto-generated training masks against GT instance annotations. Mosaic3D (Lee et al., 2025a) uses SAM2 with naive 2D-to-3D lifting; Precision and recall are computed at IoU > 0.5.

| Dataset | Masks / scene | Mean best-IoU | Prec. @0.5 | Recall @0.5 | IoU > 0.50 |
|---|---|---|---|---|---|
| Mosaic3D (2025a) | 16.1 | 0.247 | 4.8% | 2.5% | 3.8% |
| SpaCeFormer-3M (Ours) | **65.2** | **0.251** | **33.6%** | **54.3%** | **25.9%** |

MaskClustering (Yan et al., 2024) to group segments by spatial proximity, visibility, and containment. Strict filtering (visibility > 0.3, consensus > 0.9) removes incomplete fragments and yields a one-to-many mapping between each aggregated 3D mask and its constituent 2D masks, used by Step 3 for multi-view captioning.

**Step 3: Multi-view Captioning.** Single-view captioning yields view-specific descriptions rather than intrinsic properties. For each 3D instance we retrieve all views containing it (via Step 2 mapping), select up to $K$ representatives by k-medoids on frame timestamps weighted by mask visibility, and present them to the VLM (Team et al., 2025) in two formats—cropped-with-context and background-removed (Figure 3). A structured prompt (Appendix A.7) directs the VLM to describe intrinsic properties (shape, texture, material) and consistent spatial relationships. We sample multiple captions per instance by varying the view selection seed. Figure 4 shows a qualitative example.

### 3.3. Dataset Statistics

SpaCeFormer-3M contains **604,127 3D instances** from **7,361 scenes**, paired with **3.0 million captions** (Table 1). This significantly exceeds prior datasets in scale, covering both professional scans (ScanNet++) and casual captures (ARKitScenes). Captions average 15.5 words, providing sufficient detail for open-vocabulary training.

**Mask Quality Analysis.** To validate our multi-view mask clustering, we compare against Mosaic3D's (Lee et al., 2025a) SAM2-based 2D-to-3D lifting on 312 ScanNet train

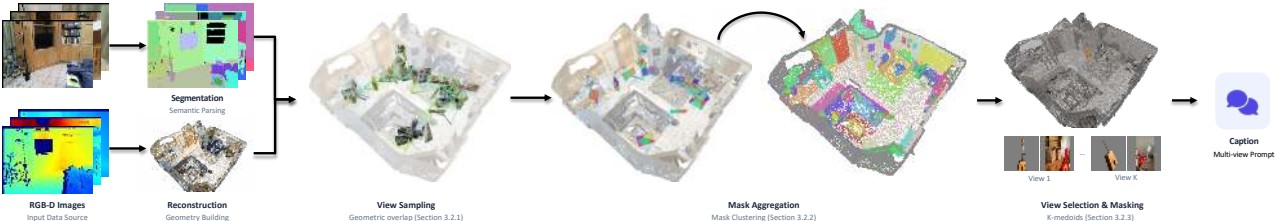

*Figure 2.* **Dataset generation pipeline.** Our pipeline generates high-quality 3D mask-caption pairs by (1) aggregating 2D masks into 3D instances through training-free multi-view clustering and (2) generating diverse, intrinsic-focused captions via structured multi-view VLM prompting. The process leverages large-scale RGB-D and video datasets to produce a massive corpus for open-vocabulary 3D learning.

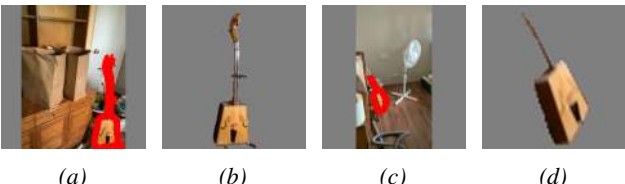

|     |     |     |     |
| --- | --- | --- | --- |
| *(a)* | *(b)* | *(c)* | *(d)* |

*Figure 3.* **Multi-view caption prompt for LLM.** To generate high-fidelity captions, we select representative views of a 3D instance and provide them to a VLM in two formats: (a, c) cropped with spatial context and (b, d) background-removed masked views. See Appendix A.7.2 for details.

scenes (Table 2). Mosaic3D produces only 16.1 masks per scene (vs. 32.0 GT instances), with 2.5% recall at IoU $> 0.5$—the vast majority of objects are missed or fragmented. SpaCeFormer-3M generates 65.2 masks per scene with **54.3% recall** and **33.6% precision**, confirming that multi-view aggregation produces substantially more complete, geometry-consistent instances.

# 4. SpaCeFormer: Space-Curve Transformer

SpaCeFormer pairs a sparse 3D transformer backbone (Sec. 4–4.3) with a proposal-free instance decoder (Sec. 4.4). The backbone is designed to serve the decoder: it produces spatially coherent, instance-aligned voxel features and a CLIP-aligned feature map consumed directly via mask attention, without external region proposals.

## 4.1. Preliminaries

Given a point cloud $\mathcal{P} = \{(p_i, f_i)\}_{i=1}^{M}$ with $p_i \in \mathbb{R}^3$ and $f_i \in \mathbb{R}^{d_{in}}$, we voxelize via average pooling into a sparse grid of $N$ non-empty voxels with features $\mathbf{X} \in \mathbb{R}^{N \times d_{in}}$. We distinguish *spatial window size* (3D geometric extent, $H \times W \times D$ voxels) from *attention window size* (token count $L$ in one attention computation).

**Space-Curve Attention Strategy.** We combine two complementary mechanisms. (i) *Spatial (window) attention*: voxels partitioned into fixed *spatial windows*; due to 3D sparsity, each window holds a variable number of voxels (1

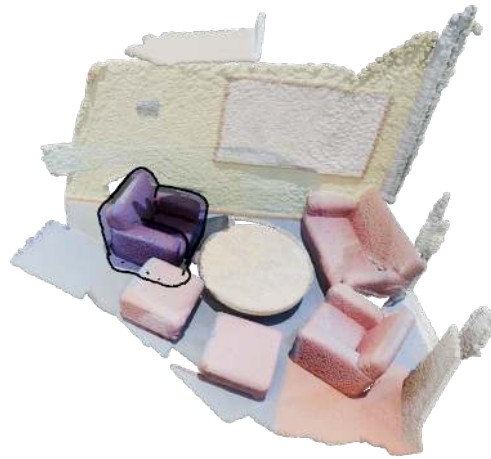

*(1) "The armchair features a smooth, red leather upholstery, providing a comfortable seating option."*
*(2) "A medium-sized, red armchair with a boxy shape sits adjacent to a round wooden table."*
*(3) "This red armchair offers a place to relax; its sturdy frame suggests durability and frequent use."*
*(4) "The armchair's design blends classic lines with a deep red hue, complementing the yellow wall."*
*(5) "Positioned near a table, the red armchair provides a functional and visually appealing seating area."*

*Figure 4.* **Office/lounge scene example.** Red leather armchair from an office waiting area.

to $H \times W \times D$), strictly preserving local geometry. (ii) *Curve (Morton) attention*: voxels serialized by Morton codes $\pi$ and partitioned into fixed-length segments of size $L$ (Wu et al., 2024), giving fixed attention window size but variable spatial extent and structured diversity. Both use full bidirectional multi-head attention within each window, reducing complexity from $O(N^2)$ to $O(N \cdot L_{\max})$. Fig. 5 contrasts the two; details in Sec. 4.3.

## 4.2. Architecture

The backbone $f_\theta$ follows a U-Net architecture with $N_{\text{blocks}}$ transformer blocks (Fig. 7). The input voxel features are first embedded into a hidden dimension $d$ as $\mathbf{H}^0 = \mathbf{X}\mathbf{W}_E$.

**Convolution Block.** Each block first extracts local features via a sparse 3D convolution with a residual shortcut:

$$\mathbf{C}^l = \text{Conv3D}(\mathbf{H}^{l-1}) + \text{Conv}_{\text{shortcut}}(\mathbf{H}^{l-1}), \quad (1)$$

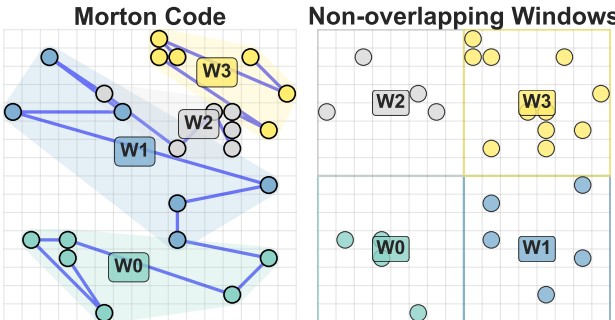

Figure 5. **Comparison of attention approaches.** (Left) Morton serialized attention showing the serialization path and resulting clusters. (Right) Non-overlapping window attention showing improved preservation of local geometric relationships. Quantitatively, our window-based approach reduces average within-window pairwise distance by 28.6% compared to Morton attention (visual coherence; see Sec. 4.3).

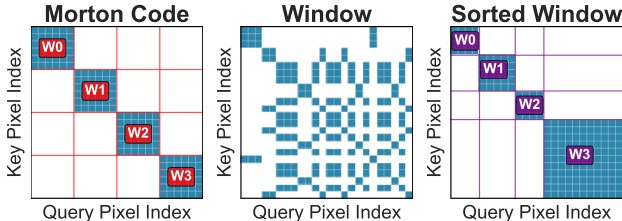

Figure 6. **Attention pattern comparison.** (Left) Morton code attention mask showing uniform block diagonal structure. (Middle) Window attention mask showing variable-sized blocks based on spatial proximity. (Right) Sorted window attention mask where pixels are reordered by window index to recover a block diagonal structure while preserving spatial relationships. Red, green, and purple lines indicate Morton, spatial, and sorted window boundaries, respectively.

injecting local geometric inductive biases before attention.

**Transformer Block.** We then apply windowed multi-head self-attention (MHSA) and a feed-forward network (FFN) with pre-LayerNorm (LN) and residual connections:

$$\mathbf{A}^l = \text{MHSA}(\text{LN}(\mathbf{C}^l)) + \mathbf{C}^l, \tag{2}$$

$$\mathbf{H}^l = \text{FFN}(\text{LN}(\mathbf{A}^l)) + \mathbf{A}^l, \tag{3}$$

where MHSA operates on the serialized windows from the preliminaries. These blocks are arranged in $N_{\text{levels}}$ encoder stages with strided-convolution downsampling and a symmetric decoder that upsamples and merges skip connections.

**Output Representation.** The final voxel features are mapped back to the original points $\mathcal{P}$ via nearest-neighbor interpolation to produce per-point features $\mathbf{F} \in \mathbb{R}^{M \times D}$ for the RoPE-enhanced decoder (Sec. 4.4), and a lightweight projector produces CLIP-aligned features $\mathbf{Z} \in \mathbb{R}^{N \times D_{\text{CLIP}}}$. To prevent overfitting to a fixed serialization, each forward pass randomly permutes the serialization strategy (Wu et al., 2024) and shifts spatial windows by $W/2$ along random axis combinations (Appendix A.5.3).

### 4.3. Space-Curve Attention: Window + Morton Code

PTv3 (Wu et al., 2024) relies solely on Morton-ordered segments, which inject beneficial randomness but sacrifice strict spatial locality by splitting spatially proximate voxels across fixed-length segments. Our space-curve attention restores locality at fine resolutions by interleaving spatial windows (fixed geometric extent, variable voxel count) with Morton segments (fixed token count, variable extent), producing sharp, instance-aligned features that the proposal-free decoder consumes directly.

We quantify the coherence gain via the mean pairwise voxel distance within an attention window (Appendix A.5.1).

Fig. 6 shows spatial window attention reduces this distance by 28.6% over Morton-based attention. Implementation and pseudocode: Appendix A.5.2, A.5.4.

**3D Rotary Positional Embedding (RoPE).** To encode 3D spatial relationships in our proposal-free architecture, we employ 3D rotary positional embeddings (RoPE) that directly encode position information into the attention mechanism through rotation transformations. For voxels at positions $\mathbf{p}_i = (x_i, y_i, z_i)$ and $\mathbf{p}_j = (x_j, y_j, z_j)$, the attention is computed as:

$$\langle \mathbf{q}_i, \mathbf{k}_j \rangle = \mathbf{q}_i \mathbf{R}_{\Theta, \mathbf{p}_j - \mathbf{p}_i} \mathbf{k}_j^\top, \tag{4}$$

where $\mathbf{R}_{\Theta, \Delta\mathbf{p}}$ is a block-diagonal rotation matrix parameterized by relative position $\Delta\mathbf{p} = (x_j - x_i, y_j - y_i, z_j - z_i)$:

$$\mathbf{R}_{\Theta, \Delta\mathbf{p}} = \begin{bmatrix} \mathbf{R}_x(\theta_x) & & \\ & \mathbf{R}_y(\theta_y) & \\ & & \mathbf{R}_z(\theta_z) \end{bmatrix}, \tag{5}$$

where each $\mathbf{R}_d(\theta_d)$ for $d \in \{x, y, z\}$ is a 2D rotation matrix applied to pairs of feature dimensions, and $\theta_d = \Delta p_d / \text{base}^{2m/d_h}$ for frequency index $m$. The block-diagonal structure allows independent encoding of each spatial dimension. We provide detailed formulations in Appendix A.1.

In our backbone, RoPE is applied with *relative* displacements within local sliding windows (Sec. 4.3); in the instance decoder (Sec. 4.4), RoPE is applied for query–scene cross-attention to preserve point locations.

### 4.4. Proposal-Free Instance Segmentation Decoder

Unlike multi-stage pipelines that depend on external region proposal networks (*e.g.* Mask3D (Schult et al., 2023), Segment3D (Huang et al., 2024a)) followed by per-proposal feature extraction and heuristic merging, our decoder predicts masks directly from a fixed set of learned queries via mask attention (Cheng et al., 2022), in the spirit of DETR (Carion

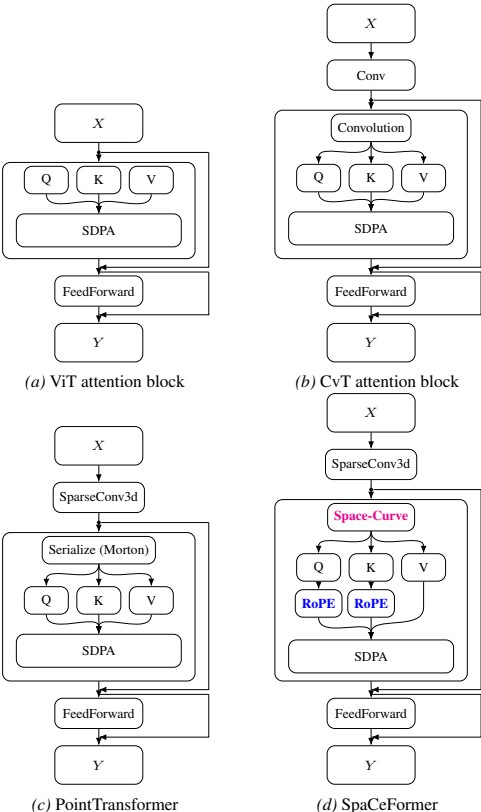

*(a)* ViT attention block     *(b)* CvT attention block

*(c)* PointTransformer     *(d)* SpaCeFormer

*Figure 7.* **Attention block diagrams.** (Left to right) Attention blocks of ViT, CvT, PointTransformer, and our SpaCeFormer. Each shows vertical flow with residuals after MHSA and FeedForward. LayerNorm, DropPath, and projection shortcuts are omitted for clarity.

et al., 2020)—end-to-end, single forward pass, no external modules or heuristic post-processing.

**Iterative Refinement with RoPE.** The decoder (Fig. 8) consumes per-point features $\mathbf{F} \in \mathbb{R}^{N \times D}$ and $Q{=}200$ learned query embeddings $\mathbf{Q}^{(0)}$. It refines queries through $T{=}3$ shared-weight iterations of cross-attention, self-attention, and FFN, using $S$ randomly sampled points per forward pass for efficiency. In cross-attention we apply 3D RoPE (Eq. (4)) to keys based on their 3D coordinates; since learned queries skip RoPE, the rotated keys encode absolute coordinates—critical for performance (Tab. 8).

**Prediction Heads.** After $T$ iterations we use all points and predict instance masks $\mathbf{M} = \sigma(\mathbf{F}(\mathbf{W}_M\mathbf{Q}^{(T)})^\top) \in \mathbb{R}^{N \times Q}$, foreground/background logits $\mathbf{C} = \mathbf{W}_C\mathbf{Q}^{(T)}$, and CLIP features $\mathbf{Z} = \mathbf{W}_{\text{CLIP}}\mathbf{Q}^{(T)}$. At inference we assign labels by argmax cosine similarity between $\mathbf{Z}$ and the text embeddings of all candidate labels in the evaluation vocabulary.

**Optimization.** We train with $\mathcal{L} = \lambda_{\text{mask}}\mathcal{L}_{\text{mask}} + \lambda_{\text{CLIP}}\mathcal{L}_{\text{CLIP}} + \lambda_{\text{fg}}\mathcal{L}_{\text{fg}}$ with Hungarian matching, where $\mathcal{L}_{\text{mask}}$ is BCE+Dice (Cheng et al., 2022), $\mathcal{L}_{\text{CLIP}}$ is InfoNCE (Oord et al., 2018), and $\mathcal{L}_{\text{fg}}$ is the query objectness loss. We use Muon (Jordan et al., 2024) for $\geq$2D hidden-layer parameters

and AdamW for the rest (Appendix A.9).

## 5. Experiments

**Datasets.** We train on SpaCeFormer-3M built via the Sec. 3 pipeline over ScanNet, ScanNet++, Matterport3D, and ARKitScenes (Dai et al., 2017; Yeshwanth et al., 2023; Chang et al., 2018; Baruch et al., 2021) (54.3% vs. 2.5% mask recall at IoU>0.5; Tab. 2). We evaluate on Replica (Straub et al., 2019) (8 scenes, zero-shot), Scan-Net++ (Yeshwanth et al., 2023) (100 classes), and Scan-Net200 (Rozenberszki et al., 2022) (200 classes).

**Implementation Details.** We train on 4 nodes $\times$ 8 NVIDIA H100 GPUs with batch size 32 for 25k iterations ($\sim$1.5 days) using Muon (Jordan et al., 2024) and AdamW. End-to-end per-scene inference latency (voxelization + backbone + decoder + mask extraction; FP16 autocast, batch 1, single NVIDIA GB200) is **0.12 s on ScanNet200, 0.22 s on Replica, and 0.30 s on ScanNet++**, tracking per-scene point counts ($\sim$170k, $\sim$800k, $\sim$1.6M avg, respectively); one-time CLIP text encoding ($\sim$0.02 s) is excluded as it amortizes across scenes. The decoder scales near-linearly in $N$ ($\propto N^{0.98}$, $R^2{=}0.97$; Fig. 9), matching the $\mathcal{O}(N \cdot Q)$ cross-attention gather over $Q{=}100$ queries; total latency scales sub-linearly ($\propto N^{0.39}$) because the sparse-conv backbone depends on voxel topology and dominates at small $N$, ceding to the decoder for $N{\gtrsim}800$k. We primarily evaluate open-vocabulary 3D *instance* segmentation.

### 5.1. Comparison with Existing Methods

**Zero-Shot Evaluation on Replica.** Replica (Straub et al., 2019) (8 scenes) is never seen at training; we use it for cross-domain transfer. Tab. 3: SpaCeFormer reaches 24.1 mAP, ahead of Open-YOLO 3D (Boudjoghra et al., 2025) (23.7) and OpenTrack3D (Zhou et al., 2025) (23.9), while using only 3D input and running 119$\times$ faster than Open-YOLO 3D and 3,900$\times$ faster than OpenMask3D in a single forward pass over the entire scene.

**Cross-Dataset Evaluation on ScanNet++.** Scan-Net++ (Yeshwanth et al., 2023) contributes 100 fine-grained classes on higher-quality reconstructions. SpaCeFormer reaches 22.9 mAP (Tab. 4), surpassing the prior best Open-Track3D (20.6) which relies on multi-view 2D inputs plus YOLO-World+SAM2 proposals, while running >4,000$\times$ faster than the 3D+2D multi-stage baselines (MaskCluster-ing 600 s, OpenMask3D $\sim$554 s).

**Evaluation on ScanNet200.** Tab. 5 categorizes ScanNet200 (200 classes) methods by input type and proposal network. Most prior work uses closed-vocabulary 3D proposals from Mask3D (Schult et al., 2023) (trained on GT); switching to open-vocabulary proposals from Segment3D (Huang et al., 2024a) collapses accuracy (Mosaic3D: 11.8→2.7

*Figure 8.* **RoPE-enhanced instance segmentation decoder architecture.** The decoder iteratively refines query tokens through cross-attention with scene features and self-attention among queries. RoPE is applied separately to encode 3D spatial relationships in both attention mechanisms using absolute coordinates from the scene.

*Table 3.* **Zero-shot 3D instance segmentation on Replica (Straub et al., 2019) (8 scenes).** All methods evaluated zero-shot (Replica is not used for training). We categorize by region proposal type: **(a)** GT-trained 3D proposals (Mask3D (Schult et al., 2023) or ISBNet (Ngo et al., 2023)). **(b)** Open-vocabulary proposals (no GT 3D annotations). **(c)** Proposal-free methods (with and without ground-truth 3D mask supervision). Baseline numbers from Open-YOLO 3D (Boudjoghra et al., 2025) and OpenTrack3D (Zhou et al., 2025). †SOLE latency estimated from its Mask3D (Schult et al., 2023) backbone; SOLE requires ScanNet200 GT mask supervision.

| | Method | Inputs | Region Proposal Network | | | mAP | $mAP_{50}$ | $mAP_{25}$ | Latency | Speedup |
| | | | Name | No GT | Open vocab | | | | | |
|---|---|---|---|---|---|---|---|---|---|---|
| (a) | OpenScene-3D (2023) | 3D | Mask3D (2023) | ✗ | ✗ | 8.2 | 10.5 | 12.6 | 4.3s | 31× |
| | OpenScene-2D (2023) | 3D + 2D | Mask3D (2023) | ✗ | ✗ | 10.9 | 15.6 | 17.3 | 317.3s | 2267× |
| | OpenMask3D (2023) | 3D + 2D | Mask3D (2023) | ✗ | ✗ | 13.1 | 18.4 | 24.2 | 547.3s | 3909× |
| | Open3DIS (2024) | 3D + 2D | ISBNet (2023) + SAM | ✗ | ✗ | 18.5 | 24.5 | 28.2 | 188.0s | 1343× |
| | Open3DIS (3D prop. only) (2024) | 3D | ISBNet (2023) | ✗ | ✗ | 14.9 | 18.8 | 23.6 | 35.1s | 251× |
| | Open-YOLO 3D (2025) | 3D + 2D | Mask3D (2023) | ✗ | ✗ | 23.7 | 28.6 | 34.8 | 16.6s | 119× |
| | Details Matter (2D) (2025) | 2D | Mask3D + GSAM | ✗ | ✗ | 20.8 | 32.4 | 38.5 | – | – |
| | Details Matter (3D) (2025) | 3D | Mask3D + GSAM | ✗ | ✗ | 22.0 | 26.7 | 32.5 | – | – |
| | Details Matter (2025) | 3D + 2D | Mask3D + GSAM | ✗ | ✗ | 22.6 | 31.7 | 37.7 | 597s | 4264× |
| (b) | OVIR-3D (2023) | 3D + 2D | Detic (2022) | ✓ | ✓ | 11.1 | 20.5 | 27.5 | 52.7s | 376× |
| | OV-MAP (2024) | 3D + 2D | SAM (2023) | ✓ | ✓ | 14.2 | 19.6 | 28.1 | – | – |
| | PoVo (2024) | 3D + 2D | SAM (2023) | ✓ | ✓ | 20.8 | 28.7 | 34.4 | – | – |
| | BoxOVIS (2025) | 3D + 2D | Box detector + SAM | ✓ | ✓ | 24.0 | 31.8 | 37.4 | 43.7s | 312× |
| | OpenTrack3D (2025) | 3D + 2D | YOLO-World + SAM2 | ✓ | ✓ | 23.9 | 36.4 | 47.6 | ∼62s | 443× |
| (c) | SOLE (2025b) | 3D | Not required (proposal-free) | ✗ | ✓ | 24.7 | 31.8 | 40.3 | ∼1.2s† | ∼5× |
| | SpaCeFormer (Ours) | 3D | Not required (proposal-free) | ✓ | ✓ | **24.1** | **31.8** | **37.1** | **0.22s** | — |

*Table 4.* **Zero-shot 3D instance segmentation on ScanNet++ (100 classes).** We compare open-vocabulary methods that assign semantic labels from text queries. All methods use the ScanNet++ validation set. †Latency estimated from ScanNet200 timing.

| | Method | Inputs | Region Proposal Network | | | mAP | $mAP_{50}$ | $mAP_{25}$ | Latency | Speedup |
| | | | Name | No GT | Open vocab | | | | | |
|---|---|---|---|---|---|---|---|---|---|---|
| (a) | OpenMask3D (2023) | 3D + 2D | Mask3D (2023) | ✗ | ✗ | 2.0 | 2.7 | 3.4 | ∼554s† | 3957× |
| | OVIR-3D (2023) | 3D + 2D | Mask3D (2023) | ✗ | ✗ | 3.6 | 5.7 | 7.3 | – | – |
| (b) | MaskClustering (2024) | 3D + 2D | CropFormer (2023) | ✓ | ✓ | 7.8 | 10.7 | 12.1 | 600s | 4286× |
| | Segment3D (2024a) | 3D + 2D | SAM (2023) | ✓ | ✓ | 10.1 | 17.7 | 20.2 | – | – |
| | Open3DIS (2024) | 3D + 2D | SAM-HQ (2023) | ✓ | ✓ | 11.9 | 18.1 | 21.7 | ∼360s† | 2571× |
| | Any3DIS (2025) | 3D + 2D | SAM2 (2024) | ✓ | ✓ | 12.9 | 19.0 | 21.9 | ∼36s† | 257× |
| | OpenSplat3D (2025) | 3D + 2D | SAM + GS | ✓ | ✓ | 16.5 | 29.7 | 39.0 | – | – |
| | OpenTrack3D (2025) | 3D + 2D | YOLO-World + SAM2 | ✓ | ✓ | 20.6 | 34.2 | 43.4 | ∼320s† | 2286× |
| (c) | SpaCeFormer (Ours) | 3D | Not required (proposal-free) | ✓ | ✓ | **22.9** | **33.7** | **41.6** | **0.30s** | — |

mAP). Our proposal-free decoder reaches 11.1 mAP without any proposals, 4.1× better than Mosaic3D+Segment3D, at 0.12 s per scene. Fully supervised closed-vocab references: Mask3D 27.4, OneFormer3D 30.6. Qualitative results in Fig. 10; novel categories (*e.g.* Snoopy, X-mas) absent from the taxonomy in appendix Fig. 15. Per-class breakdown and confusion analysis: Appendix A.14.

**Class-Agnostic Mask Quality.** Tab. 6 isolates mask quality from semantic labeling. With 3D-only input SpaCeFormer reaches 22.5 AP, beating multi-stage 3D+2D methods (MaskClustering 19.2, OnlineAnySeg 18.6), and our $AP_{50}/AP_{25}$ (45.7/64.4) clear all open-vocab baselines including Any3DIS 2D-only (45.2/55.0), confirming the proposal-free decoder produces tight boundaries.

*Table 5.* **Zero-shot 3D instance segmentation on ScanNet200 (Rozenberszki et al., 2022).** For a fair comparison, we categorize methods by input types and region proposal network: **(a)** Methods using both 3D point cloud and 2D RGB-D images, with 3D+2D region proposals and 2D CLIP inference. **(b)** Methods using both 3D+2D inputs, with region proposals from Mask3D (Schult et al., 2023) (closed-vocab) and 2D CLIP inference. **(c)** Methods using only 3D input with Mask3D (Schult et al., 2023). **(d)** Methods using only 3D input with open-vocabulary 3D region proposals. $^{\dagger}$ denotes results without test-time voting, following the official implementation. Latency reports end-to-end runtime (seconds) per scene on ScanNet validation, including both proposal generation and method inference.

| | Method | Inputs | Region Proposal Network | | | mAP | $mAP_{50}$ | $mAP_{25}$ | $mAP_{head}$ | $mAP_{com.}$ | $mAP_{tail}$ | Latency |
| | | | Name | No GT | Open vocab | | | | | | | |
|---|---|---|---|---|---|---|---|---|---|---|---|---|
| (a) | Open3DIS (2024) | 3D + 2D | Superpoints (2004) + ISBNet (2023) + Grounded-SAM (2024) | ✗ | ✗ | 23.7 | 29.4 | 32.8 | 27.8 | 21.2 | 21.8 | 33.5 |
| | Any3DIS (2025) | 3D + 2D | ISBNet (2023) + SAM2 (2024) | ✗ | ✗ | 25.8 | – | – | 27.4 | 23.8 | 26.4 | – |
| | Details Matter (2025) | 3D + 2D | Mask3D (2023) + GSAM (2024) | ✗ | ✗ | 25.8 | 32.5 | 36.2 | 26.3 | 23.2 | 28.2 | – |
| | SAI3D (2024) | 3D + 2D | Superpoints (2004) + SAM (2023) | ✓ | ✓ | 12.7 | 18.8 | 24.1 | 12.1 | 10.4 | 16.2 | 75.2 |
| | MaskClustering (2024) | 3D + 2D | CropFormer (2023) | ✓ | ✓ | 12.0 | 23.3 | 30.1 | – | – | – | – |
| | SAM2Object (2025) | 3D + 2D | SAM2 (2024) | ✓ | ✓ | 13.3 | 19.0 | 23.8 | – | – | – | – |
| | OpenTrack3D (2025) | 3D + 2D | YOLO-World + SAM2 (2024) | ✓ | ✓ | 26.0 | 37.7 | 45.4 | – | – | – | ∼356 |
| (b) | OpenScene-2D (2023) | 3D + 2D | Mask3D (2023) | ✗ | ✗ | 11.7 | 15.2 | 17.8 | 13.4 | 11.6 | 9.9 | - |
| | OpenScene-2D/3D (2023) | 3D + 2D | Mask3D (2023) | ✗ | ✗ | 5.3 | 6.7 | 8.1 | 11.0 | 3.2 | 1.1 | - |
| | OpenMask3D (2023) | 3D + 2D | Mask3D (2023) | ✗ | ✗ | 15.4 | 19.9 | 23.1 | 17.1 | 14.1 | 14.9 | 553.9 |
| | Open-YOLO 3D (2025) | 3D + 2D | Mask3D (2023) | ✗ | ✗ | 24.7 | 31.7 | 36.2 | 27.8 | 24.3 | 21.6 | 21.8 |
| (c) | OpenScene-3D (2023) | 3D | Mask3D (2023) | ✗ | ✗ | 4.8 | 6.2 | 7.2 | 10.6 | 2.6 | 0.7 | 1.1 |
| | RegionPLC (2024b) | 3D | Mask3D (2023) | ✗ | ✗ | 6.3 | 8.6 | 9.7 | 15.6 | 1.0 | 1.7 | 1.0 |
| | OpenIns3D (2024b) | 3D | Mask3D (2023) | ✗ | ✗ | 8.8 | 10.3 | 14.4 | 16.0 | 6.5 | 4.2 | 285.2 |
| | OpenIns3D$^{\dagger}$ (2024b) | 3D | Mask3D (2023) | ✗ | ✗ | 3.3 | 5.0 | 5.6 | 7.0 | 1.4 | 1.2 | 50.0 |
| | Mosaic3D (2025a) | 3D | Mask3D (2023) | ✗ | ✗ | 11.8 | 16.0 | 17.8 | 21.8 | 7.2 | 5.4 | 1.0 |
| | SpaCeFormer w/ Proposal | 3D | Mask3D (2023) | ✗ | ✗ | 16.7 | 21.9 | 24.8 | 22.8 | 16.4 | 9.9 | 1.1 |
| (d) | OpenScene-3D (2023) | 3D | Segment3D (2024a) | ✓ | ✓ | 0.6 | 1.0 | 1.6 | 1.4 | 0.4 | 0.0 | 2.0 |
| | RegionPLC (2024b) | 3D | Segment3D (2024a) | ✓ | ✓ | 1.5 | 2.1 | 2.6 | 2.3 | 0.2 | 1.9 | 1.9 |
| | OpenIns3D$^{\dagger}$ (2024b) | 3D | Segment3D (2024a) | ✓ | ✓ | 1.7 | 2.7 | 3.7 | 3.2 | 0.8 | 1.0 | 64.8 |
| | Mosaic3D (2025a) | 3D | Segment3D (2024a) | ✓ | ✓ | 2.7 | 4.2 | 5.7 | 3.8 | 2.0 | 2.4 | 1.9 |
| | Mosaic3D w/ Decoder | 3D | Not required (proposal-free) | ✓ | ✓ | 3.9 | 7.0 | 12.3 | 6.6 | 2.1 | 2.8 | 1.2 |
| | SpaCeFormer | 3D | Not required (proposal-free) | ✓ | ✓ | **11.1** | **18.8** | **24.3** | **13.2** | **9.5** | **10.6** | **0.12** |

*Table 6.* **Class-agnostic 3D instance segmentation on Scan-Net200.** We report mask-quality metrics (AP, $AP_{50}$, $AP_{25}$) without semantic labels, evaluated on ScanNet200 validation (198 foreground classes). Methods are grouped by input modality. Latency is end-to-end seconds per scene. $^{*}$ uses closed-vocabulary ISBNet proposals trained with ground-truth annotations.

| Method | Inputs | AP | $AP_{50}$ | $AP_{25}$ | Latency |
|---|---|---|---|---|---|
| *3D + 2D methods (use multi-view RGB-D at inference)* | | | | | |
| OVIR-3D (2023) | 3D + 2D | 14.4 | 27.5 | 38.8 | 466.8 |
| MaskClustering (2024) | 3D + 2D | 19.2 | 36.6 | 51.7 | 156.0 |
| Any3DIS$^{*}$ (2025) | 3D + 2D | **42.5** | **51.2** | 54.5 | ∼36 |
| *2D-only methods* | | | | | |
| Any3DIS (2025) | 2D | **32.5** | **45.2** | 55.0 | ∼36 |
| *3D-only methods (point cloud only at inference)* | | | | | |
| SpaCeFormer (Ours) | 3D | 22.5 | 45.7 | 64.4 | 0.2 |

*Table 7.* **Space-curve attention ablation.** Comparing window attention only, Morton-curve ordering only, and their strategic combination. Evaluated on ScanNet200 zero-shot instance segmentation validation set.

| Name | Window | Morton | Class Agnostic | | | 200-Way Class Specific | | |
| | | | AP | AP25 | AP50 | mAP | mAP25 | mAP50 |
|---|---|---|---|---|---|---|---|---|
| SpaCeFormer | ✓ | – | **0.25466** | 0.68991 | 0.49903 | 0.09517 | 0.22476 | 0.16815 |
| SpaCeFormer | – | ✓ | 0.23311 | 0.67391 | 0.47290 | 0.09470 | 0.22315 | 0.16410 |
| SpaCeFormer | ✓ | ✓ | 0.25178 | **0.69114** | **0.50447** | **0.11092** | **0.24312** | **0.18783** |

*Table 8.* Ablation of positional encoding (PE) strategy. We test absolute positional embedding (APE), learnable relative positional bias (Bias), rotary positional embedding (RoPE), and no positional encoding (No PE).

| Method | mAP | $mAP_{50}$ | $mAP_{25}$ | $mAP_{head}$ | $mAP_{com.}$ | $mAP_{tail}$ |
|---|---|---|---|---|---|---|
| SpaCeFormer (No PE) | 5.97 | 11.47 | 17.27 | 9.68 | 4.50 | 3.43 |
| SpaCeFormer (APE) | 5.95 | 11.54 | 16.10 | 9.35 | 5.37 | 2.68 |
| SpaCeFormer (Bias) | 6.46 | 11.82 | 17.22 | 9.20 | 5.15 | 4.84 |
| SpaCeFormer (RoPE) | **7.60** | **13.73** | **18.57** | **10.62** | **6.47** | **5.42** |

## 5.2. Ablation Study

Unless noted, all ablations train Space-Curve Transformer-512C for 25k iterations with Muon (lr $2e-3$), local batch size 2 on 32 H100 GPUs, and report zero-shot performance on ScanNet200 validation.

**Space-Curve Attention.** Tab. 7 compares Morton-only (PTv3), window-only, and our combined space-curve attention; the combination wins, confirming the two mechanisms are complementary rather than redundant.

**Positional Encoding Strategy.** RoPE in the decoder yields 7.60 mAP, beating no-PE (5.97), absolute PE (5.95), and positional bias (6.46); see Tab. 8.

**Query Initialization Strategy.** Learned queries (DETR-style) reach 19.82 class-agnostic AP, beating FPS (17.97) and random (14.64); see Tab. 9.

**Hyperparameter Choices.** Defaults: random key sampling with a training-only cap, default normalization placement, and window attention at the two highest-resolution levels (sizes 64 and 48). Full ablations: Appendix A.11.

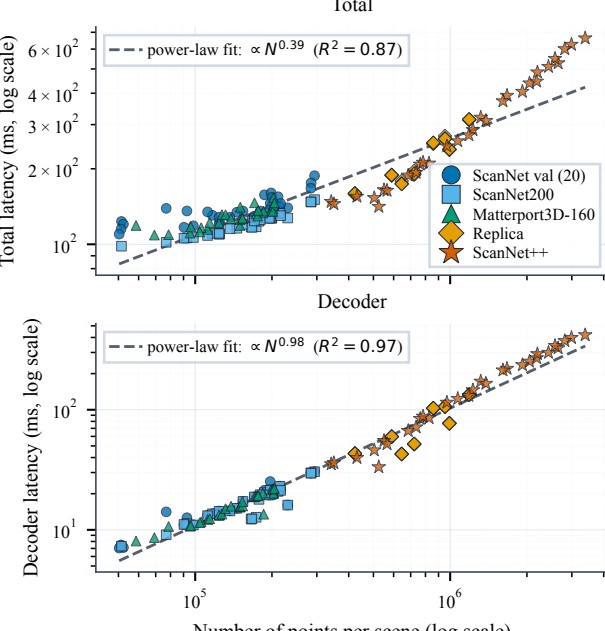

Figure 9. **Per-stage inference latency vs. scene point count.** Total end-to-end (top) and decoder-only (bottom) latency across 160 scenes from the five zero-shot eval splits.

Table 9. Ablation of query initialization strategy. We compare 100 learned queries, Farthest Point Sampling (FPS), and random initialization. AP, $AP_{25}$, $AP_{50}$ are class-agnostic; remaining columns are class-wise mAP.

| Name | Class Agnostic | | | 200-Way Class Specific | | |
|---|---|---|---|---|---|---|
| | AP | AP25 | AP50 | mAP | mAP25 | mAP50 |
| SpaCeFormer (Learned) | **0.19819** | **0.62469** | **0.41561** | 0.06443 | **0.18213** | 0.12037 |
| SpaCeFormer (FPS) | 0.17970 | 0.58387 | 0.38380 | **0.07164** | 0.17942 | **0.13114** |
| SpaCeFormer (Random) | 0.14635 | 0.52839 | 0.32332 | 0.05634 | 0.14893 | 0.10230 |

**Backbone Comparison.** Under matched training, Space-Curve Transformer beats PTv3 on all four benchmarks (*e.g.* 16.55 vs. 14.77 mIoU on ScanNet200), showing spatial windows add value over Morton-only serialization. Full analysis: Appendix A.12.

## 6. Conclusion

SpaCeFormer pairs (1) SpaCeFormer-3M, a large-scale dataset with 3.0M multi-view-consistent captions over 604K geometry-consistent instances from training-free multi-view mask clustering plus multi-view captioning, with (2) space-curve attention that combines spatial windows and Morton curve serialization to produce locally coherent features, consumed by a RoPE-enhanced proposal-free decoder. At **0.22 s per scene on Replica** (0.12 s on ScanNet200, 0.30 s on ScanNet++)—2–3 orders of magnitude faster than multi-stage 2D+3D pipelines—SpaCeFormer reaches **24.1 zero-shot mAP on Replica** (Pareto-optimal among open-vocab methods without GT 3D supervision; Fig. 1), 22.9 mAP on ScanNet++ (best among all reported methods, including 2D+3D), and 11.1 mAP on ScanNet200 under its matched

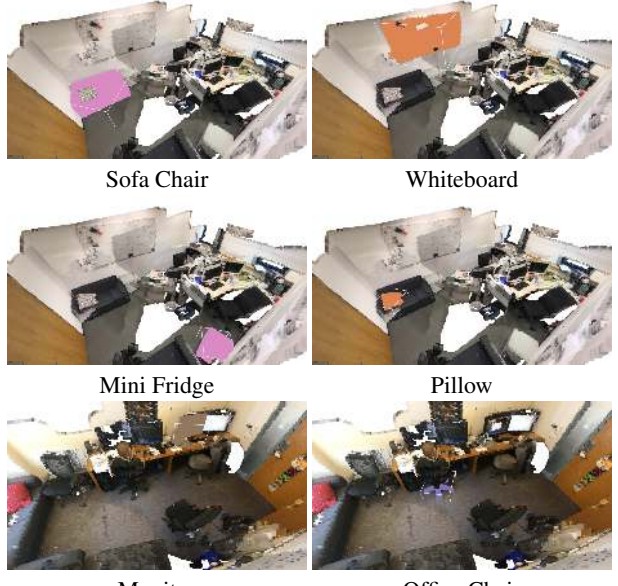

Figure 10. **Qualitative results on ScanNet200.** We visualize the 3D masks predicted by our model. The results demonstrate the ability to segment a wide range of objects, including furniture like sofa chairs, mini fridges, monitors, and office chairs.

proposal-free, 3D-only, no-GT-3D setting.

**Limitations and Future Work.** A gap remains between our 3D-only proposal-free setup and methods using multi-view 2D inputs or GT-trained proposals on ScanNet200 (*e.g.* OpenTrack3D 26.0 mAP with 2D+3D, Open-YOLO 3D 24.7 mAP with Mask3D proposals); closing it without sacrificing speed—for instance, by distilling 2D proposal knowledge into the 3D backbone—is a promising direction. The fixed number of learned queries ($Q$=200) can also cap recall on cluttered scenes with many small objects, which adaptive or hierarchical query mechanisms may alleviate. Finally, our evaluation is restricted to indoor scenes; extending SpaCeFormer-3M and SpaCeFormer to outdoor and in-the-wild settings remains an important direction for future work, especially for robotic perception.

**Acknowledgements.** This work was partly supported by IITP grants (RS-2024-00457882: National AI Research Lab Project, RS-2022-II220290: Visual Intelligence for Space-Time Understanding and Generation) funded by the Korea government (MSIT).

## Impact Statement

This paper advances open-vocabulary 3D instance segmentation; lower-latency 3D perception benefits robotics and AR/VR while reducing compute cost. SpaCeFormer-3M uses public indoor benchmarks with VLM-generated captions, which may inherit model biases. We see no other specific ethical risks.

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

# A. Appendix

## A.1. 3D Rotary Positional Embeddings (RoPE) - Detailed Mathematics

In this section, we provide a detailed mathematical exposition of the 3D Rotary Positional Embeddings (RoPE) used in our voxel transformer architecture. RoPE directly encodes relative positional information through rotation transformations applied to query and key vectors.

### A.1.1. BACKGROUND: 2D ROTARY EMBEDDINGS

The original RoPE formulation (Su et al., 2024) was designed for 1D sequences in language models. The key insight is to encode position through rotation of query and key vectors in the complex plane. For a 2D vector $\mathbf{x} = [x_1, x_2]^\top$, a rotation by angle $\theta$ can be expressed as:

$$\begin{bmatrix} x'_1 \\ x'_2 \end{bmatrix} = \begin{bmatrix} \cos\theta & -\sin\theta \\ \sin\theta & \cos\theta \end{bmatrix} \begin{bmatrix} x_1 \\ x_2 \end{bmatrix} \tag{6}$$

Alternatively, treating the vector as a complex number $z = x_1 + ix_2$, rotation becomes multiplication by $e^{i\theta}$:

$$z' = z \cdot e^{i\theta} = (x_1 + ix_2)(\cos\theta + i\sin\theta) \tag{7}$$

### A.1.2. EXTENSION TO 3D VOXEL SPACE

For 3D voxel data, we extend RoPE to encode positions along three spatial dimensions. The attention score between voxels at positions $\mathbf{p}_i = (x_i, y_i, z_i)$ and $\mathbf{p}_j = (x_j, y_j, z_j)$ is computed as:

$$\langle \mathbf{q}_i, \mathbf{k}_j \rangle = \mathbf{q}_i \mathbf{R}_{\Theta, \mathbf{p}_j - \mathbf{p}_i} \mathbf{k}_j^\top, \tag{8}$$

where $\mathbf{R}_{\Theta, \Delta\mathbf{p}}$ is a block-diagonal rotation matrix parameterized by the relative position $\Delta\mathbf{p} = (x_j - x_i, y_j - y_i, z_j - z_i)$.

**Rotation Matrix Structure.** The rotation matrix $\mathbf{R}_{\Theta, \Delta\mathbf{p}}$ has a block-diagonal structure:

$$\mathbf{R}_{\Theta, \Delta\mathbf{p}} = \begin{bmatrix} \mathbf{R}_x(\theta_x) & & \\ & \mathbf{R}_y(\theta_y) & \\ & & \mathbf{R}_z(\theta_z) \end{bmatrix}, \tag{9}$$

where each $\mathbf{R}_d(\theta_d)$ for $d \in \{x, y, z\}$ consists of multiple 2D rotation blocks applied to pairs of feature dimensions.

**Frequency Computation.** For feature dimension $d_h$ (head dimension), we allocate dimensions equally among the three spatial axes. Each spatial dimension uses $d_{\text{rope}}/3$ dimensions where:

$$d_{\text{rope}} = \left\lfloor \frac{d_h}{6} \right\rfloor \times 6 \tag{10}$$

The rotation angles are computed using position-dependent frequencies:

$$\theta_d^{(m)} = \frac{\Delta p_d}{\text{base}^{2m/(d_{\text{rope}}/3)}}, \quad m \in \{0, 1, \ldots, d_{\text{rope}}/6 - 1\} \tag{11}$$

where base (typically $10^4$) controls the wavelength of the sinusoidal embeddings, and $\Delta p_d$ is the relative position difference along dimension $d$.

### A.1.3. APPLICATION TO QUERY AND KEY VECTORS

Given query and key vectors $\mathbf{q}, \mathbf{k} \in \mathbb{R}^{d_h}$, the rotation matrix $\mathbf{R}_{\Theta, \Delta\mathbf{p}}$ operates on pairs of dimensions. Specifically, each $\mathbf{R}_d(\theta_d)$ for $d \in \{x, y, z\}$ is composed of $d_{\text{rope}}/6$ independent 2D rotation matrices:

$$\mathbf{R}_d(\theta_d) = \text{diag}\left(\mathbf{R}_d^{(0)}, \mathbf{R}_d^{(1)}, \ldots, \mathbf{R}_d^{(d_{\text{rope}}/6-1)}\right), \tag{12}$$

where each 2D rotation block is:

$$\mathbf{R}_d^{(m)} = \begin{bmatrix} \cos\theta_d^{(m)} & -\sin\theta_d^{(m)} \\ \sin\theta_d^{(m)} & \cos\theta_d^{(m)} \end{bmatrix} \tag{13}$$

The complete transformation can be written as:

$$\mathbf{q}' = \mathbf{R}_{\Theta,\Delta\mathbf{P}}\mathbf{q} = \begin{bmatrix} \mathbf{R}_x(\theta_x) & & & \\ & \mathbf{R}_y(\theta_y) & & \\ & & \mathbf{R}_z(\theta_z) & \\ & & & \mathbf{I}_{d_h - d_{\text{rope}}} \end{bmatrix} \mathbf{q} \tag{14}$$

where $\mathbf{I}_{d_h - d_{\text{rope}}}$ is an identity matrix for dimensions not affected by RoPE.

### A.1.4. PROPERTIES OF 3D RoPE

**Relative Position Encoding.** The key property of RoPE is that the attention score between two voxels depends only on their relative position:

$$\langle \mathbf{q}_i, \mathbf{k}_j \rangle = \mathbf{q}_i \mathbf{R}_{\Theta,\mathbf{p}_j - \mathbf{p}_i} \mathbf{k}_j^\top = f(\mathbf{q}_i, \mathbf{k}_j, \Delta\mathbf{p}_{ij}) \tag{15}$$

where $\Delta\mathbf{p}_{ij} = \mathbf{p}_j - \mathbf{p}_i = (x_j - x_i, y_j - y_i, z_j - z_i)$ is the relative position vector.

**Translation Invariance.** The attention scores are invariant to global translations. If all voxel positions are shifted by $\mathbf{t} = (t_x, t_y, t_z)$:

$$\langle \mathbf{q}_i, \mathbf{k}_j \rangle_{\mathbf{p}+\mathbf{t}} = \langle \mathbf{q}_i, \mathbf{k}_j \rangle_{\mathbf{p}} \tag{16}$$

This property ensures that the model learns spatial patterns independent of absolute position in the scene.

**Long-Range Decay.** The positional encoding naturally decays for distant voxels due to the periodic nature of the sinusoidal functions, helping the model focus on local spatial relationships while maintaining awareness of global structure.

### A.1.5. IMPLEMENTATION DETAILS

In practice, our implementation includes several optimizations:

1. **On-the-fly computation**: Unlike language models that can cache positional embeddings for fixed sequence positions, we compute RoPE dynamically for each voxel's coordinates, as sparse voxel data can have arbitrary coordinate values.

2. **Coordinate normalization**: To handle large coordinate values, we normalize by subtracting the minimum coordinate in each batch:

$$\tilde{x}_i = x_i - \min_j(x_j) + 1 \tag{17}$$

3. **Learnable frequencies**: Optionally, the frequency parameters $\theta_j$ can be made learnable:

$$\theta_j = \text{softplus}(\mathbf{w}_j)/\text{base}^{2j/(d_{\text{rope}}/3)} \tag{18}$$

where $\mathbf{w}_j$ are learnable parameters.

### A.1.6. COMPARISON WITH ALTERNATIVE POSITION ENCODINGS

Table 10 compares 3D RoPE with alternative positional encoding methods for voxel data:

The key advantages of 3D RoPE are:

- **No additional parameters**: Position information is encoded through rotation, not learned embeddings

- **Better extrapolation**: Can handle coordinate ranges not seen during training

- **Computational efficiency**: No need to store large positional embedding tables

- **Natural 3D structure**: Separately encodes x, y, z dimensions while maintaining their relationships

*Table 10.* Comparison of positional encoding methods for 3D voxel transformers

| Method | Translation Invariant | Memory Efficient | Extrapolation |
|---|:---:|:---:|:---:|
| Additive sinusoidal PE | ✗ | ✓ | ✗ |
| Learnable PE | ✗ | ✗ | ✗ |
| Relative PE | ✓ | ✗ | ✓ |
| 3D RoPE (ours) | ✓ | ✓ | ✓ |

## A.2. Window Attention with RoPE

When applying RoPE within windowed attention, special care must be taken to preserve relative position information. For each window $\mathcal{W}_i$ containing voxels with positions $\{\mathbf{p}_1, \mathbf{p}_2, \ldots, \mathbf{p}_{|\mathcal{W}_i|}\}$:

1. **Global coordinate preservation**: We use the original voxel positions, not window-local indices. The attention computation within window $\mathcal{W}_i$ is:

$$\langle \mathbf{q}_m, \mathbf{k}_n \rangle = \mathbf{q}_m \mathbf{R}_{\Theta, \mathbf{p}_n - \mathbf{p}_m} \mathbf{k}_n^\top \quad \forall v_m, v_n \in \mathcal{W}_i \tag{19}$$

2. **Cross-window consistency**: The rotation matrix $\mathbf{R}_{\Theta, \Delta\mathbf{p}}$ depends only on the relative position $\Delta\mathbf{p}$, ensuring consistent encoding regardless of window assignment.

3. **Window-aware normalization**: When normalizing coordinates to handle large values, we use global statistics:

$$\tilde{\mathbf{p}}_j = \mathbf{p}_j - \min_{k \in \text{batch}} (\mathbf{p}_k) + \mathbf{1} \tag{20}$$

where $\mathbf{1} = (1, 1, 1)$ ensures non-zero positions.

This design ensures that relative positions between voxels are preserved even when they are assigned to different windows, maintaining the translation invariance property of RoPE.

## A.3. RoPE in Instance Segmentation Decoder: Complete Formulation

This section provides the complete mathematical formulation of RoPE in the instance segmentation decoder (Sec. 4.4), including detailed cross-attention equations, self-attention formulations, and analysis of the benefits.

### A.3.1. DETAILED CROSS-ATTENTION FORMULATION

At iteration $t$, we randomly sample $S$ points from the scene to form keys and values. Let $\mathcal{S}_t \subseteq \{1, \ldots, N\}$ denote the sampled indices with corresponding CLIP features $\mathbf{F}_{\text{CLIP},s} = \{\mathbf{f}_{s_j}^{\text{clip}}\}_{j=1}^S$ and coordinates $\mathbf{P}_s = \{\mathbf{p}_{s_j}\}_{j=1}^S$. For cross-attention, we apply RoPE rotations $\mathcal{R}(\mathbf{x}, \mathbf{p})$ to both queries and keys based on their 3D coordinates:

$$\widetilde{\mathbf{Q}}^{(t)} = \{\mathcal{R}(\mathbf{W}_Q \mathbf{q}_i^{(t)}, \mathbf{p}_{q,i})\}_{i=1}^Q, \tag{21}$$

$$\widetilde{\mathbf{K}}_s = \{\mathcal{R}(\mathbf{W}_K \mathbf{f}_{s_j}^{\text{clip}}, \mathbf{p}_{s_j})\}_{j=1}^S, \tag{22}$$

$$\mathbf{V}_s = \mathbf{W}_V \mathbf{F}_{\text{CLIP},s}, \tag{23}$$

$$\mathbf{A}^{(t)} = \text{Softmax}\left( \frac{\widetilde{\mathbf{Q}}^{(t)} \widetilde{\mathbf{K}}_s^\top}{\sqrt{d_h}} + \mathbf{M}^{(t)} \right), \tag{24}$$

$$\mathbf{Q}^{(t+0.5)} = \mathbf{A}^{(t)} \mathbf{V}_s, \tag{25}$$

where $\mathbf{W}_Q, \mathbf{W}_K, \mathbf{W}_V$ are learned projection matrices, $d_h$ is the head dimension, and $\mathbf{M}^{(t)} \in \{0, -\infty\}^{Q \times S}$ is an attention mask computed from predicted instance masks to focus queries on relevant regions.

**Key Difference from Backbone RoPE.** In the backbone (Sec. 4.3), RoPE encodes the relative displacement $\mathbf{p}_j - \mathbf{p}_i$ between voxels within a local window. However, in the decoder, queries represent *instance proposals* at specific spatial locations, and we need to preserve their absolute positions to distinguish between instances at different locations. Therefore,

we apply RoPE using absolute coordinates—rotating queries and keys independently by $\mathcal{R}(\cdot, \mathbf{p}_i)$ and $\mathcal{R}(\cdot, \mathbf{p}_j)$ respectively. Due to rotation matrix properties, this still captures relative positions while preserving absolute spatial information in the feature representations.

### A.3.2. SELF-ATTENTION WITH ROPE

After cross-attention, queries interact through self-attention to model relationships between instance proposals. We apply RoPE to the self-attention queries and keys using the fixed query coordinates $\mathbf{P}_q$:

$$\widetilde{\mathbf{Q}}_{\text{self}}^{(t)} = \{\mathcal{R}(\mathbf{W}_Q^s \mathbf{q}_i^{(t+0.5)}, \mathbf{p}_{q,i})\}_{i=1}^Q, \tag{26}$$

$$\widetilde{\mathbf{K}}_{\text{self}}^{(t)} = \{\mathcal{R}(\mathbf{W}_K^s \mathbf{q}_i^{(t+0.5)}, \mathbf{p}_{q,i})\}_{i=1}^Q, \tag{27}$$

$$\mathbf{Q}^{(t+0.75)} = \text{SelfAttn}(\widetilde{\mathbf{Q}}_{\text{self}}^{(t)}, \widetilde{\mathbf{K}}_{\text{self}}^{(t)}), \tag{28}$$

where $\mathbf{W}_Q^s, \mathbf{W}_K^s$ are self-attention projection matrices. This allows queries to be spatially aware of their locations when interacting, enabling the model to learn that nearby queries should cooperate (e.g., for parts of the same object) while distant queries can specialize independently.

### A.3.3. COMPLETE ITERATIVE REFINEMENT

After self-attention, a feed-forward network further processes the queries:

$$\mathbf{Q}^{(t+1)} = \text{FFN}(\mathbf{Q}^{(t+0.75)}) + \mathbf{Q}^{(t+0.75)}. \tag{29}$$

This process repeats for $T$ iterations (typically $T = 3$ in our experiments) with shared weights across iterations, allowing progressive refinement of instance proposals.

### A.3.4. BENEFITS OF ROPE IN INSTANCE SEGMENTATION

Integrating RoPE into the instance segmentation decoder provides several advantages:

**(1) Spatial Awareness:** RoPE enables queries to be spatially aware during both cross-attention (when attending to scene features) and self-attention (when interacting with other queries). This is crucial for instance segmentation, where the spatial location of a proposal determines which scene regions it should attend to.

**(2) Absolute Position Encoding:** Unlike relative position encoding in the backbone, absolute position encoding in the decoder allows the model to distinguish between instances at different spatial locations, preventing ambiguity when multiple instances of the same category exist.

**(3) Geometry-Aware Reasoning:** By encoding 3D coordinates directly into the attention mechanism, RoPE helps the model reason about geometric relationships—for example, queries at similar heights are more likely to belong to similar object categories (tables, chairs) than queries at very different heights (floor, ceiling).

**(4) Extrapolation:** RoPE's continuous nature allows the model to generalize to scenes larger than those seen during training, as the rotation angles smoothly vary with coordinate values.

### A.4. Attention Normalization

The placement of layer normalization (LN) within attention blocks is a critical design choice that affects training stability and performance. We consider three variants: Default, PreNorm, and PostNorm architectures, which differ in both the placement of normalization and the source of residual connections.

**Default architecture.** In the default variant, layer normalization is applied to the input before it branches into attention or MLP operations, and the residual connections use the normalized values:

$$\widetilde{\mathbf{C}}^l = \text{LN}(\mathbf{C}^l), \tag{30}$$

$$\mathbf{A}^l = \text{MHSA}(\widetilde{\mathbf{C}}^l) + \widetilde{\mathbf{C}}^l, \tag{31}$$

$$\widetilde{\mathbf{A}}^l = \text{LN}(\mathbf{A}^l), \tag{32}$$

$$\mathbf{H}^l = \text{FFN}(\widetilde{\mathbf{A}}^l) + \widetilde{\mathbf{A}}^l, \tag{33}$$

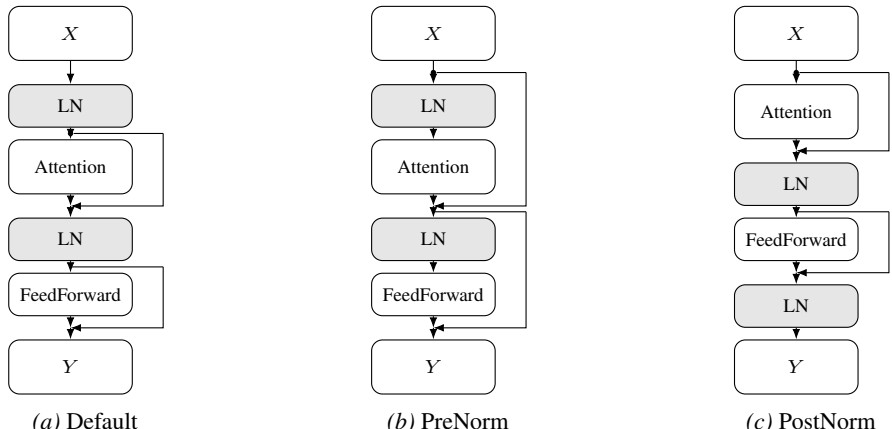

*Figure 11.* **Attention normalization variants.** (Left to right) Default, PreNorm, and PostNorm attention block architectures. Default applies LayerNorm (LN) before branching into attention/MLP, with residuals from normalized values. PreNorm applies LN inline before operations, with residuals from original values. PostNorm applies LN after operations and residual additions. Residual connections are shown with skip paths.

where $\mathbf{C}^l$ is the convolved feature from the previous layer, MHSA is multi-head self-attention, FFN is the feed-forward network, and LN denotes layer normalization. This variant applies normalization before branching, ensuring both the operation and residual use normalized inputs.

**PreNorm architecture.** In the PreNorm variant, layer normalization is applied inline before the operations, but the residual connections use the original (unnormalized) values:

$$\mathbf{A}^l = \text{MHSA}(\text{LN}(\mathbf{C}^l)) + \mathbf{C}^l, \tag{34}$$

$$\mathbf{H}^l = \text{FFN}(\text{LN}(\mathbf{A}^l)) + \mathbf{A}^l. \tag{35}$$

This is the standard PreNorm architecture used in modern transformers such as Vision Transformer (ViT) and is known for its training stability, especially in deep networks. The key difference from the default variant is that residuals come from the original values before normalization.

**PostNorm architecture.** In the PostNorm variant, layer normalization is applied after the attention and feed-forward operations and their residual additions:

$$\mathbf{A}^l = \text{LN}(\text{MHSA}(\mathbf{C}^l) + \mathbf{C}^l), \tag{36}$$

$$\mathbf{H}^l = \text{LN}(\text{FFN}(\mathbf{A}^l) + \mathbf{A}^l). \tag{37}$$

The PostNorm architecture was used in the original Transformer paper and can provide different gradient flow characteristics, though it may require more careful initialization and learning rate scheduling for deep networks.

**Implementation.** Our framework supports all three variants through the `AttentionBlock`, `AttentionBlockPreNorm`, and `AttentionBlockPostNorm` classes, allowing for empirical comparison of normalization strategies in 3D vision transformers. We apply these designs consistently in both backbone and decoder blocks. The PreNorm variant is used as the default in our experiments due to its superior training stability, which we found particularly important for end-to-end training of our *proposal-free* instance segmentation model.

### A.5. Additional Method Details

A.5.1. SPATIAL COHERENCE METRIC

To quantify the spatial coherence improvement, we measure the average pairwise distance between voxels within the same attention window. For a window $\mathcal{W}$ containing $n$ voxels, the spatial coherence is defined as:

$$C(\mathcal{W}) = \frac{1}{\binom{n}{2}} \sum_{i=1}^{n} \sum_{j=i+1}^{n} \|\mathbf{p}_i - \mathbf{p}_j\|_2, \tag{38}$$

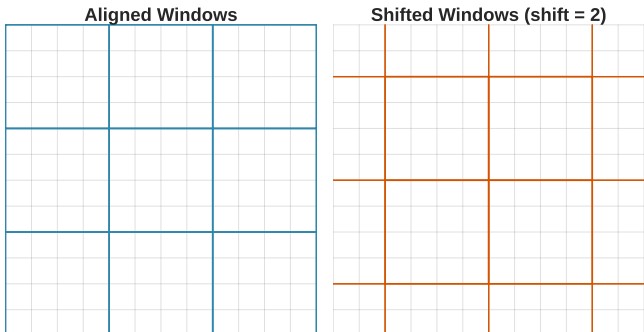

*Figure 12.* **SWIN windowing on a 2D grid.** (Left) Non-overlapping windows aligned to the grid. (Right) Non-overlapping windows shifted by half the window size, illustrating SWIN's shifted windowing.

where $\mathbf{p}_i = (x_i, y_i, z_i)$ and $\mathbf{p}_j = (x_j, y_j, z_j)$ are the 3D coordinates of voxels $v_i$ and $v_j$ in window $\mathcal{W}$. Lower values indicate better spatial coherence.

### A.5.2. WINDOW ATTENTION IMPLEMENTATION

We operate spatial window attention in 3D cubic windows of shape $H \times W \times D$ voxels. For each window, we compute per-head queries, keys, and values only on the active voxels inside the cube. A window mask $\mathbf{M}$ restricts interactions to within the cube, ignoring padded or empty positions. Morton code attention uses fixed-length segments as described in (Wu et al., 2024). Both mechanisms scan the entire space while maintaining local interactions.

### A.5.3. SHIFTED WINDOW ATTENTION

To enable cross-window connections and prevent overfitting to a fixed grid, we adopt a randomized shifted windowing scheme. Unlike the deterministic alternating shifts in Swin Transformer (Liu et al., 2021), we randomly sample a shift configuration for each layer. The window partition can be shifted by half the window size ($W/2$) along any combination of axes: x, y, z, xy, xz, yz, or xyz. Formally, for each axis $d \in \{x, y, z\}$, the shift is independently chosen as either $0$ or $W_d/2$.

Formally, for sliding windows with stride $s$ and window dimensions $(W_x, W_y, W_z)$, we define:

$$
\begin{aligned}
\mathcal{W}^{\text{sliding}}_{(a,b,c)} = \{v_i : & as \leqslant x_i < as + W_x, \\
& bs \leqslant y_i < bs + W_y, \\
& cs \leqslant z_i < cs + W_z\},
\end{aligned}
\tag{39}
$$

where $(a, b, c)$ are indices along each axis. This ensures voxels in the same spatial neighborhood are processed together.

### A.5.4. SPACE-CURVE ATTENTION PSEUDOCODE

Algorithm 1 summarizes our space-curve attention strategy, which combines spatial window attention and Morton curve attention at each encoder/decoder level.

### A.6. Dataset Generation Details

**Data Processing Details.** We implement standardized preprocessing modules for the major indoor scene datasets.

- **ScanNet**: We parse the proprietary `.sens` format using a custom SensorData reader, extracting JPEG-compressed color frames and zlib-compressed depth maps at two-frame intervals. Camera poses are derived from the sensor's camera-to-world matrices, and intrinsics are scaled according to the target resolution (default: 640px maximum dimension).

- **ARKitScenes**: We process trajectory files containing angle-axis pose representations, converting them to 4×4 transformation matrices. Device orientation changes are handled by detecting rotations and applying appropriate image rotations (0°, 90°, 180°, 270°) using OpenCV transforms.

---

**Algorithm 1** Space-Curve Attention

---

**Require:** Voxel features $\mathbf{H} \in \mathbb{R}^{N \times d}$, coords $\mathbf{P} \in \mathbb{R}^{N \times 3}$, level $\ell$
1: **if** level $\ell$ uses spatial windows **then**
2:      Sample random shift $\delta \in \{0, W_\ell/2\}^3$
3:      $\{\mathcal{W}_k\} \leftarrow \text{SPATIALPARTITION}(\mathbf{P} + \delta, W_\ell)$
4:      **for** each window $\mathcal{W}_k$ **do**
5:          $\mathbf{Q}, \mathbf{K}, \mathbf{V} \leftarrow \text{project}(\mathbf{H}[\mathcal{W}_k])$
6:          Apply 3D RoPE to $\mathbf{Q}, \mathbf{K}$ with relative positions
7:          $\mathbf{H}[\mathcal{W}_k] \leftarrow \text{MHSA}(\mathbf{Q}, \mathbf{K}, \mathbf{V})$
8:      **end for**
9: **else**
10:      $\pi \leftarrow \text{MORTONSERIALIZE}(\mathbf{P})$
11:      Partition $\pi$ into fixed-length segments of size $L$
12:      **for** each segment $\mathcal{S}_k$ **do**
13:          $\mathbf{Q}, \mathbf{K}, \mathbf{V} \leftarrow \text{project}(\mathbf{H}[\mathcal{S}_k])$
14:          Apply 3D RoPE to $\mathbf{Q}, \mathbf{K}$ with relative positions
15:          $\mathbf{H}[\mathcal{S}_k] \leftarrow \text{MHSA}(\mathbf{Q}, \mathbf{K}, \mathbf{V})$
16:      **end for**
17: **end if**
18: **return** $\mathbf{H}$

---

- **ScanNet++**: We utilize COLMAP reconstructions, parsing `images.txt` to extract quaternion-based poses and converting them to world-to-camera matrices. We integrate with NeRFStudio-format transforms to extract intrinsics from `transforms_undistorted.json`.

- **Matterport3D**: We apply spatial filtering based on point cloud bounding boxes to select relevant viewpoints. Separate pose and intrinsic files are converted from Matterport's coordinates to our standardized format while maintaining a depth scale factor of 4000.0.

**Overlap-based View Sampling.** We compute overlap between consecutive frames via backproject depth to obtain per-frame point clouds.

$$\bigcap(\mathcal{P}_{t_i}, \mathcal{P}_{t_j}) = \frac{1}{|\mathcal{P}_{t_i}|} \sum_{p \in \mathcal{P}_{t_i}} \mathbb{1}\left[\min_{q \in \mathcal{P}_{t_j}} \|p - q\|_2 < \varepsilon\right], \tag{40}$$

where $\mathcal{P}_{t_i} \in \mathbb{R}^{HW \times 3}$ is the backprojected per-frame point cloud at timestep $t_i$, $\mathbb{1}[\cdot]$ is the indicator function, and $\varepsilon$ is a 3D proximity threshold.

**View Selection for Captioning.** We select up to $K$ representative views using weighted k-medoids clustering over the temporal dimension. The distance matrix is $D_{jk} = |t_j - t_k|$ for frame timestamps, with weights $w_j = \sum_{p \in M_i^j} \mathbb{1}[p \in M_i^j]$ corresponding to mask pixel count in each view. This weighting scheme prioritizes frames with high instance visibility while maintaining temporal spread, ensuring informative viewpoints for caption generation.

**Quality Control.** The slight missing caption files (13 out of 15,880 total) in ScanNet++ and Matterport3D represent scenes that failed post-processing quality checks due to insufficient multi-view support or depth quality issues. These exclusions constitute less than 0.1% of the total data and do not affect overall dataset integrity. All remaining mask-caption pairs pass our geometric consistency and multi-view agreement thresholds established in the aggregation pipeline.

### A.7. Multi-view Captioning Details

In this section, we provide the full system prompt, user prompt, and additional examples of the multi-view captioning process described in Sec. 3.

### A.7.1. SYSTEM PROMPT

The following system prompt is used to instruct the VLM (Gemma 3) to generate intrinsic and spatially consistent captions:

---

**Full System Prompt**

You are an expert image-captioning assistant. You will be given multiple "views" of the same subject. Your job is to synthesize these views into one cohesive description that focuses on the subject's intrinsic properties. Note that this "object" might actually be a wall, floor, or part of a larger structure—so it may not always be a standalone, traditional object. In such cases, describe the subject as best you can (color, texture, material, shape, typical size, etc.):

**Object Properties**

- Emphasize color, shape, material, texture, and typical size (e.g., small, medium, large).

- If uncertain about the object type, use best judgment (e.g., "a piece of fabric" could be "a pillow" if it looks stuffed).

**Relationships to Other Objects**

- Only mention other objects if they appear consistently across all views.

- At the end of your description, state how the foreground object is related spatially to these other objects (e.g., "It is on a table with a blue vase and a green plant").

- Do not mention the camera's or viewer's position (e.g., "top-left corner," "on the right side").

**What Not to Mention**

- Do not mention bounding boxes, outlines, image padding, or differences between views (e.g., "In the first image...").

- Do not describe masked cutouts or the act of masking—they are merely visualization aids.

**Output Format**

- Provide a single concise paragraph that directly describes the object.

- Avoid filler phrases such as "The object is..." or "In the image...".

- End with a short sentence describing its spatial relationship to other objects (if any).

Example: "A medium-sized red ceramic vase with a smooth, glossy finish. It has a narrow neck and a rounded body. It is on a table with a blue vase and a green plant."
Note: Each image is labeled "View N: ...". Any images sharing the same N come from the same original viewpoint (e.g., different mask types or outlines for the same photo).

---

### A.7.2. USER PROMPT AND EXAMPLE

We allow the user prompt to handle the multi-view logic. Below is the user prompt template and a sample assistant response.

---

**User Prompt**

Describe the object shown across all views and its relationship to other objects in the scene. Do not describe location relative to the viewpoint or image as the captions will be used for 3D datasets which will combine all camera views. Some object may be located within another object, such as a bag in a cabinet. Disregard the background and focus on the foreground object. View 1: A cropped view of the object with a red outline drawn around it: `<IMAGE_1>`. View 1: A cutout/mask that isolates the foreground object and hides the background: `<IMAGE_2>`. View 2: A cropped view of the object with a red outline drawn around it: `<IMAGE_3>`. View 2: A cutout/mask that isolates the foreground object and hides the background: `<IMAGE_4>`.

---

**Assistant Caption**

A tall, slender musical instrument with a light brown, wooden body and neck. It features a rectangular sound box with decorative carvings and a long, curved neck extending upwards. It is situated on a wooden surface near a fan and a cabinet.

---

### A.7.3. ADDITIONAL EXAMPLES

Figure 3 in the main text illustrates the pipeline for a musical instrument.

### A.8. Additional Dataset Examples

In this section, we provide additional examples of 3D instance masks and their corresponding captions from our SpaCeFormer-3M dataset. These examples demonstrate the diversity of scenes, objects, and caption quality across different environments

including office/lounge areas, communal waiting rooms, bedrooms, and retail spaces.

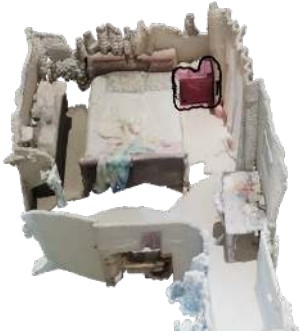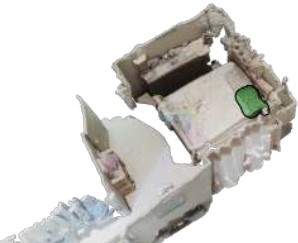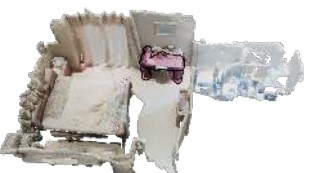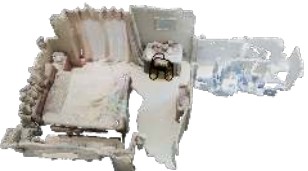

*(1) "A small, dark wood nightstand provides storage near the bed, featuring brass-toned hardware and a traditional design."*
*(2) "The reddish-brown bedside table sits adjacent to the bed, its rectangular shape complementing the room's layout."*
*(3) "This medium-sized wooden cabinet offers a surface for lamps and books, easily accessible from the bed."*
*(4) "Crafted from wood with a visible grain, the nightstand's design blends classic style with functional storage."*
*(5) "The dark brown nightstand, positioned next to the bed, provides a convenient spot for personal items and a lamp."*

*(1) "The pillow's woven fabric offers a soft, comfortable surface for resting."*
*(2) "A rectangular pillow with a muted brown and black checkered pattern rests against the bed."*
*(3) "Medium-sized pillow, suitable for supporting the head during sleep or relaxation."*
*(4) "The pillow's design complements the bedroom's decor, providing a cozy accent."*
*(5) "Positioned against the headboard, the pillow offers support and adds visual texture to the bed."*

*(1) "Dark brown wooden table with ornate carved details, providing a surface for small items."*
*(2) "Rectangular table with a glossy finish, positioned near a curtain and a chair."*
*(3) "Medium-sized table offering a flat surface for holding stationery and decorative objects."*
*(4) "The table's design features cabriole legs and a drawer, blending into the room's decor."*
*(5) "A dark wood table sits adjacent to a chair, providing a functional space for writing or display."*

*(1) "The chair's dark fabric upholstery provides a comfortable seating surface, designed for supporting a person."*
*(2) "A medium-sized chair with a curved backrest sits adjacent to a dark wooden desk, displaying a classic design."*
*(3) "This chair offers a place to sit; its slender frame allows for easy movement around the room, suitable for quick tasks."*
*(4) "The chair's metal legs and dark fabric seat create a simple, functional design, blending into the room's decor."*
*(5) "Positioned near a desk, the chair's dark color contrasts with the lighter carpet, offering a spot for focused work."*

*Figure 13.* **Bedroom scene examples.** Nightstand, pillow, ornate table, desk chair.

## A.9. Detailed Training Objectives

In this section, we provide the detailed mathematical formulations for the training objectives summarized in Sec. 4.4.

### A.9.1. HUNGARIAN MATCHING COST

Let $\{\mathbf{m}_q\}_{q=1}^{Q}$ be the predicted instance masks (columns of $\mathbf{M} \in \mathbb{R}^{N \times Q}$) and $\{\hat{\mathbf{m}}_k\}_{k=1}^{K}$ the ground-truth instance masks. We assign ground-truth instances to queries via bipartite matching using the Hungarian algorithm. The matching cost $\mathcal{C}(q, k)$ between query $q$ and ground truth $k$ is a weighted sum of classification, mask, and Dice costs (Cheng et al., 2022):

$$\mathcal{C}(q,k) \;=\; \lambda_{\text{class}}\, \mathcal{C}_{\text{class}}(q,k) \;+\; \lambda_{\text{mask}}\, \mathcal{L}_{\text{bce}}(\mathbf{m}_q, \hat{\mathbf{m}}_k) \;+\; \lambda_{\text{dice}}\, (1 - \text{Dice}(\mathbf{m}_q, \hat{\mathbf{m}}_k)).$$

Here $\mathcal{C}_{\text{class}}$ uses the query foreground/background probabilities, and $\mathcal{L}_{\text{bce}}$ denotes the sigmoid binary cross-entropy loss.

### A.9.2. LOSS COMPONENTS

**Mask Loss.** For matched pairs $(q, k) \in \mathcal{M}$, we optimize a combination of sigmoid BCE and Dice loss on the per-point mask probabilities:

$$\mathcal{L}_{\text{mask}} \;=\; \sum_{(q,k) \in \mathcal{M}} \left[ \lambda_{\text{bce}}\, \mathcal{L}_{\text{bce}}(\mathbf{m}_q, \hat{\mathbf{m}}_k) + \lambda_{\text{dice}}\, \mathcal{L}_{\text{dice}}(\mathbf{m}_q, \hat{\mathbf{m}}_k) \right].$$

where $\mathcal{L}_{\text{dice}}(\mathbf{m}, \hat{\mathbf{m}}) = 1 - \frac{2\sum_i m_i \hat{m}_i}{\sum_i m_i + \sum_i \hat{m}_i}$ is the Dice loss.

**CLIP Alignment (Contrastive).** We align the predicted CLIP features $\mathbf{Z}_q$ (obtained via a linear projection $\mathbf{Z}_q = \mathbf{W}_{\text{CLIP}}\mathbf{Q}^{(T)}$) to target CLIP embeddings $\{\mathbf{t}_k\}$ via an InfoNCE objective (Oord et al., 2018). We compute the alignment probability $p(k|q)$ using a softmax over cosine similarities:

$$p(k|q) = \frac{\exp(\cos(\mathbf{Z}_q, \mathbf{t}_k)/\tau)}{\sum_{k'} \exp(\cos(\mathbf{Z}_q, \mathbf{t}_{k'})/\tau)}.$$

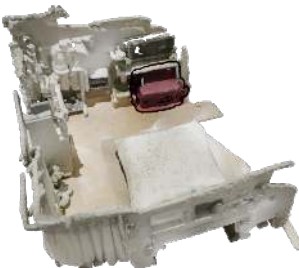 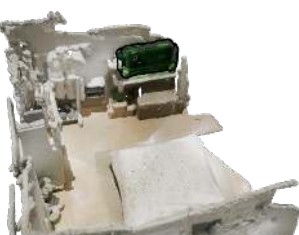 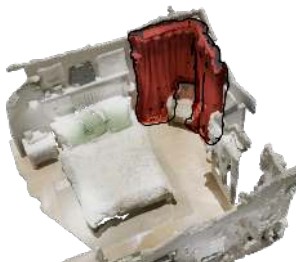 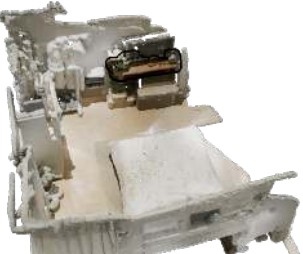

*(1) "Woven basket with a rustic texture, ideal for storing small items or displaying decorative objects."*
*(2) "Brown rectangular box with a dark strap, situated near a wooden crate and potted plant."*
*(3) "Medium-sized storage box, offering a place to keep belongings organized and easily accessible."*
*(4) "Handwoven wicker construction, blending seamlessly with the room's natural aesthetic and wooden flooring."*
*(5) "Dark brown basket, positioned beside a wooden crate, providing a functional and decorative element."*

*(1) "Dark, rectangular shelf made of composite material, providing storage space above a bench."*
*(2) "Black shelf with a flat surface, positioned on a wall near clothing racks and a plant."*
*(3) "Small, wall-mounted shelf offering a place to display decorative items or small objects."*
*(4) "The shelf's simple, angular design complements the minimalist interior decor, offering a clean aesthetic."*
*(5) "A dark shelf, slightly angled, sits above a bench, creating a layered display area within the retail space."*

*(1) "Cream-colored curtains with a soft, flowing texture, likely made of linen or a similar fabric, hang near a window."*
*(2) "The light beige curtains form a vertical drape, positioned beside a wicker chair and a potted plant."*
*(3) "These medium-length curtains offer privacy and diffuse light, suitable for a bedroom or living space."*
*(4) "The curtains' simple design complements the room's decor, creating a calming and inviting atmosphere."*
*(5) "The fabric drapes gracefully, partially obscuring a window and providing a soft backdrop to the room's furnishings."*

*(1) "The shelf is constructed from dark brown wood, providing a surface for displaying items."*
*(2) "A medium-sized, rectangular shelf sits beneath a hanging rack, its gray finish blending with the room's decor."*
*(3) "This shelf offers a small space for storage or display, easily accessible near the clothing rack."*
*(4) "The shelf's simple, linear design complements the minimalist aesthetic of the bedroom setting, offering a practical surface."*
*(5) "Positioned next to a woven basket, the dark gray shelf provides a stable platform for decorative objects and small items."*

*Figure 14.* **Retail/bedroom scene examples.** Woven basket, wall-mounted shelf, curtains, storage shelf.

We then maximize the similarity of matched pairs by minimizing the negative log-likelihood:

$$\mathcal{L}_{\text{CLIP}} = -\sum_{(q,k) \in \mathcal{M}} \log p(k|q).$$

Note that this differs from masked pooling approaches; we directly learn to predict CLIP features from the query embeddings.

**Query Foregroundness.** We supervise the query objectness scores $\mathbf{C} \in \mathbb{R}^{Q \times 2}$ with a binary label indicating whether a query is matched to any ground truth, using a 2-way cross-entropy loss:

$$\mathcal{L}_{\text{fg}} = \text{CE}(\mathbf{C}, \mathbf{y}_{\text{fg}}).$$

**Optional Closed-Set Semantic Head.** When enabled, we apply cross-entropy over the backbone per-voxel logits $\mathbf{Y}$ with ground-truth semantic labels $\hat{\mathbf{y}}$:

$$\mathcal{L}_{\text{sem}} = \text{CE}(\mathbf{Y}, \hat{\mathbf{y}}).$$

### A.10. Muon Optimization Details

Muon performs standard SGD with momentum to obtain an update matrix $G$, and then post-processes $G$ with a Newton–Schulz iteration that approximately orthogonalizes the update (i.e., replaces $G$ by the nearest semi-orthogonal matrix) before applying it to the parameters. This orthogonalization is implemented efficiently with a quintic Newton–Schulz map and runs stably in bfloat16 on modern GPUs.

Let $g_t$ be the gradient of a 2D parameter (or a batch of 2D parameters) at step $t$. Muon maintains a momentum buffer $m_t$ and forms an SGD–Nesterov update

$$m_t = \beta\, m_{t-1} + (1 - \beta)\, g_t, \qquad G_t = (1 - \beta)\, g_t + \beta\, m_t, \tag{41}$$

where $G_t$ has shape $n \times m$ along its last two dimensions. Muon then approximately orthogonalizes $G_t$ via a Newton–Schulz (NS) iteration. Define the initialization

$$X_0 = \frac{G_t}{\|G_t\|_F + \varepsilon}, \tag{42}$$

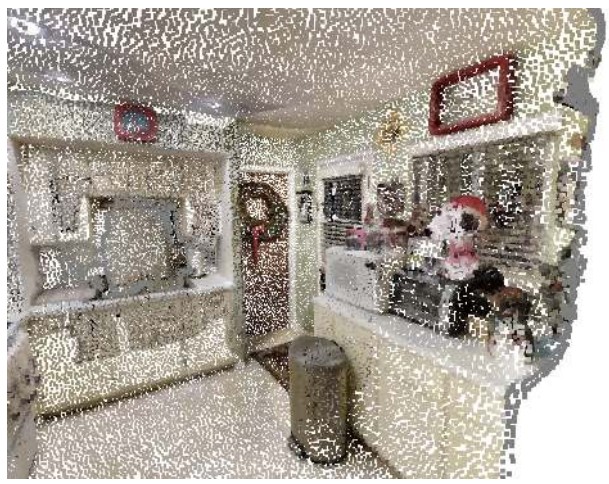 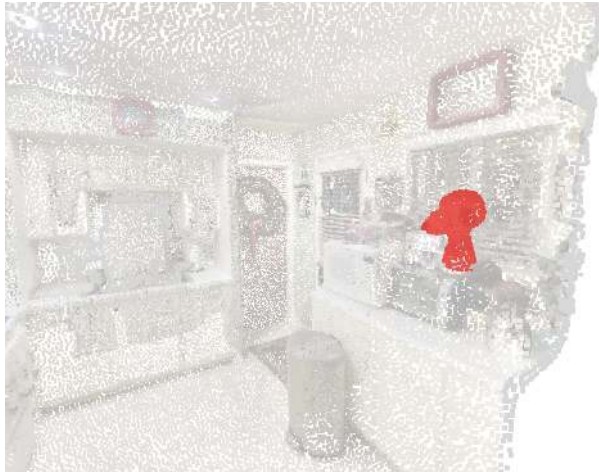

Snoopy

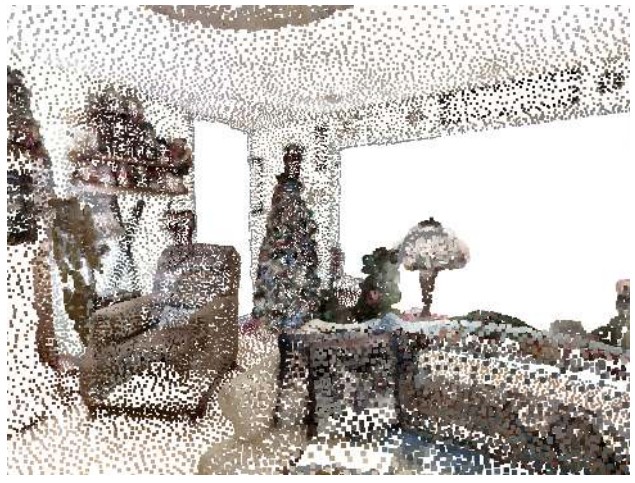 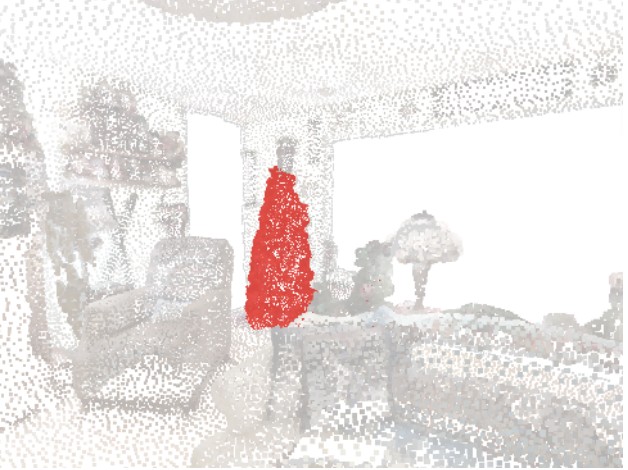

X-mas

*Figure 15.* **Novel category segmentation on Matterport3D.** We visualize predictions on categories absent from the ScanNet200 taxonomy. Left: input point cloud. Right: predicted mask (red). The results demonstrate that our model generalizes to novel objects unseen during training, such as a toy and a Christmas tree.

and iterate for $k = 0, \ldots, N - 1$ (with $N = 5$ in our experiments):

$$A_k = X_k X_k^\top, \tag{43}$$

$$X_{k+1} = aX_k + (bA_k + cA_k^2)X_k. \tag{44}$$

With coefficients $(a, b, c) = (3.4445, -4.7750, 2.0315)$, this quintic NS map drives the singular values toward $\approx 1$ and yields an update close to the nearest semi-orthogonal matrix to $G_t$ (Jordan et al., 2024). Denoting the SVD $G_t = USV^\top$ and the elementwise polynomial $\varphi(x) = ax + bx^3 + cx^5$, one step produces $U\varphi(S)V^\top$; repeated composition $\varphi^N$ sharpens singular values toward unity. The implementation uses bfloat16 arithmetic and batched matrix operations.

Finally, we apply a shape correction factor to balance tall versus wide matrices and weight decay/learning-rate scaling when updating the parameter. Our implementation function is `zeropower_via_newtonschulz`, which applies the above NS iteration to the last two dimensions (supporting batched inputs) before the parameter update. Algorithm 2 summarizes the orthogonalization routine and Algorithm 3 shows a per-parameter Muon update.

**Batched parameterization for Muon.** Muon operates on 2D matrices along their last two dimensions and naturally supports batched matrices via leading dimensions. To make convolutional kernels weights compatible while leveraging this batching, we re-parameterize them as follows:

- **Convolutions**: we arrange each 3D kernel into a tensor of shape $K \times C_{\text{in}} \times C_{\text{out}}$, where $K$ is the product of spatial kernel

---

**Algorithm 2** ZeroPower via Newton–Schulz

---

**Require:** $G \in \mathbb{R}^{\cdots \times n \times m}$, steps $N$, $(a, b, c)$, $\varepsilon > 0$

1: $X \leftarrow G$; if $n > m$ then $X \leftarrow X^\top$
2: $X \leftarrow X/(\|X\|_F + \varepsilon)$
3: **for** $k = 1..N$ **do**
4: $\quad A \leftarrow XX^\top$; $\quad X \leftarrow aX + (bA + cA^2)X$
5: **end for**
6:
7: **return** $X$ if $n \leq m$ else $X^\top$

---

---

**Algorithm 3** Muon step for one parameter

---

**Require:** $W$, gradient $g$, momentum $m$, lr $\eta$, decay $\lambda$, $\beta$, steps $N$

1: $m \leftarrow \beta m + (1-\beta)g$
2: $G \leftarrow (1-\beta)g + \beta m$
3: View last two dims as matrix: conv $\rightarrow K \times C_{in} \times C_{out}$; MoE $\rightarrow E \times C_1 \times C_2$
4: $\tilde{G} \leftarrow$ ZeroPower-NS$(G, N)$
5: $\tilde{G} \leftarrow \tilde{G} \cdot \sqrt{\max(1, n/m)}$
6: $W \leftarrow (1-\eta\lambda)W$
7: $W \leftarrow W - \eta\tilde{G}$

---

sizes (e.g., $k_x k_y k_z$). This yields a batch of $K$ matrices of size $C_{\text{in}} \times C_{\text{out}}$ on which Muon applies its orthogonalized momentum update in parallel.

These batched matrix layouts are critical for Muon: they expose the correct 2D structure to the optimizer without flattening across unrelated axes, preserve per-kernel/per-expert update geometry, and allow efficient fused batched Newton–Schulz steps. In practice, we use 5 Newton–Schulz iterations with Nesterov momentum for the internal SGD, following (Jordan et al., 2024).

### A.11. Hyperparameter Ablations

In this section, we provide detailed ablation studies on hyperparameter choices that were explored during model development. While these experiments informed our final configuration, they represent implementation details rather than core conceptual contributions. Unless noted, all ablations train Space-Curve Transformer-512C for 25k iterations with Muon (lr 2e−3), local batch size 2 on 32 H100 GPUs, and evaluate zero-shot 3D instance segmentation on ScanNet200 validation.

#### A.11.1. SAMPLING STRATEGY

Table 11 analyzes different key sampling strategies for the decoder. We find that **Random** sampling outperforms Full sampling, and applying a cap only during training (**Train Only**) yields better results than always capping. Furthermore, updating samples once per **Epoch** (forward pass) proves superior to re-sampling at every decoder step.

#### A.11.2. NORMALIZATION STRATEGY

We evaluate three normalization placement strategies: default normalization (as defined in Section A.4), pre-attention normalization, and post-attention normalization. Table 12 shows that post-attention normalization performs slightly better across all benchmarks, though differences are modest (within 0.5% mIoU). We use default normalization in our final model as it provides a good balance of performance and training stability.

#### A.11.3. WINDOW SIZE ANALYSIS

We further analyze which hierarchical levels should use window attention versus point-based Morton ordering. Table 13 shows results for different configurations. The configuration **Enc. + Dec. 64/48** where the highest and the second highest resolution layers of window size 64 and 48 respectively achieved the best performance, demonstrating that applying window attention at the finest hierarchical levels while using Morton ordering at coarser levels optimally balances local spatial locality with global scene structure.

*Table 11.* Ablation of key sampling configurations. **Strategy** denotes point selection method (Full: all points, Random: random subset). **Cap Mode** controls when to limit sample size (Always: fixed limit, Train Only: limit during training only). **Update Mode** determines sampling frequency (Step: resample every decoder layer, Epoch: sample once per forward pass).

| Strategy | Cap Mode | Update Mode | Class Agnostic | | | 200-Way Class Specific | | |
|---|---|---|---|---|---|---|---|---|
| | | | AP | AP25 | AP50 | mAP | mAP25 | mAP50 |
| Full | Always | Step | 0.24039 | 0.67884 | 0.48610 | 0.08925 | 0.20947 | 0.15192 |
| Full | Always | Epoch | 0.23490 | 0.65596 | 0.46941 | 0.09867 | 0.22479 | 0.17256 |
| Full | Train Only | Step | 0.23384 | 0.65098 | 0.45943 | 0.09332 | 0.21709 | 0.16726 |
| Full | Train Only | Epoch | 0.23592 | 0.66030 | 0.47371 | 0.09167 | 0.22452 | 0.16596 |
| Random | Always | Step | 0.24028 | 0.66436 | 0.48141 | 0.09276 | 0.21519 | 0.16254 |
| Random | Always | Epoch | 0.24256 | **0.68503** | 0.48837 | 0.09418 | 0.22070 | 0.16201 |
| Random | Train Only | Step | 0.23211 | 0.65906 | 0.47518 | 0.09217 | 0.21420 | 0.16125 |
| Random | Train Only | Epoch | **0.24615** | 0.68180 | **0.49060** | **0.10007** | **0.23347** | **0.17483** |

*Table 12.* Ablation of normalization strategy. See Section 11 for definitions.

| Method | mAP | $mAP_{50}$ | $mAP_{25}$ | $mAP_{head}$ | $mAP_{com.}$ | $mAP_{tail}$ |
|---|---|---|---|---|---|---|
| SpaCeFormer (Default) | 7.78 | 14.24 | 18.93 | 10.58 | 7.14 | 5.28 |
| SpaCeFormer (PreNorm) | 7.60 | 13.73 | 18.57 | 10.62 | 6.47 | 5.42 |
| SpaCeFormer (PostNorm) | 6.23 | 12.34 | 17.43 | 9.45 | 5.21 | 3.70 |

## A.12. Backbone Comparison and Configuration Analysis

We compare our Space-Curve Transformer backbone against the PTv3 baseline across four benchmarks, and analyze the effect of hidden dimension and window size configurations. Table 14 shows that Space-Curve Transformer consistently outperforms PTv3 across all benchmarks under identical training conditions (same optimizer, learning rate, batch size of 32 on 16 GPUs, 25k iterations). Increasing hidden dimension and adding spatial window attention at fine-grained levels further improves performance.

## A.13. Class-Agnostic Instance Segmentation

Table 15 reports class-agnostic AP (mask quality independent of semantic labeling) on ScanNet++ and Replica. On ScanNet++, SpaCeFormer achieves 29.8 AP, outperforming MaskClustering (27.9 AP) despite using 3D-only input and no external proposals. More notably, our $AP_{50}$ (54.8) and $AP_{25}$ (75.1) substantially exceed all baselines, and our recall (74.7) nearly doubles that of Any3DIS (40.8). On Replica (zero-shot), SpaCeFormer achieves 33.2 AP with 69.8 recall, demonstrating strong cross-domain mask quality generalization.

## A.14. Per-Class Confusion Matrix

We visualize the row-normalized confusion matrix (recall) for the best performing model in Figure 22. The diagonal dominance indicates strong performance across most categories, though some confusion persists between semantically similar classes such as table/desk and cabinet/shelves.

*Table 13.* Ablation of window sizes. We vary the window size configuration for the Encoder only, Decoder only, or both (Enc. + Dec.). When two window sizes are denoted (e.g., 64 / 48), they correspond to the finest (L1) and second finest (L2) hierarchies, respectively.

| Module | Win. Size | Class Agnostic | | | 200-Way Class Specific | | |
|---|---|---|---|---|---|---|---|
| | | AP | AP25 | AP50 | mAP | mAP25 | mAP50 |
| Encoder | 32 | 0.23785 | 0.68051 | **0.48933** | **0.10504** | 0.23717 | **0.18500** |
| Encoder | 64 | 0.24273 | **0.68163** | 0.48702 | 0.09958 | 0.23536 | 0.17870 |
| Encoder | 128 | 0.23956 | 0.67103 | 0.48083 | 0.09648 | 0.22886 | 0.17206 |
| Encoder | 32 / 24 | 0.23904 | 0.67107 | 0.48086 | 0.10010 | **0.23914** | 0.17849 |
| Encoder | 64 / 48 | **0.24629** | 0.67780 | 0.48662 | 0.09657 | 0.23014 | 0.16899 |
| Encoder | 128 / 96 | 0.23392 | 0.67035 | 0.47444 | 0.09183 | 0.21528 | 0.16142 |
| Decoder | 32 | 0.24228 | 0.67949 | 0.48448 | 0.09612 | 0.22739 | 0.16881 |
| Decoder | 64 | **0.24655** | 0.68339 | 0.49284 | 0.09064 | 0.21653 | 0.16108 |
| Decoder | 128 | 0.24228 | **0.68729** | **0.49285** | 0.09999 | **0.22970** | 0.17504 |
| Decoder | 32 / 24 | 0.24299 | 0.67593 | 0.48244 | 0.09331 | 0.22171 | 0.16277 |
| Decoder | 64 / 48 | 0.24575 | 0.68258 | 0.48697 | **0.10213** | 0.22904 | **0.17516** |
| Decoder | 128 / 96 | 0.23770 | 0.67711 | 0.48474 | 0.09404 | 0.22491 | 0.17398 |
| Enc. + Dec. | 32 | 0.23624 | 0.66382 | 0.47006 | 0.09703 | 0.22197 | 0.16720 |
| Enc. + Dec. | 32 / 24 | 0.23908 | 0.66986 | 0.47967 | 0.10649 | 0.23082 | 0.18304 |
| Enc. + Dec. | 128 | **0.25190** | 0.68012 | 0.49562 | 0.09169 | 0.21653 | 0.16040 |
| Enc. + Dec. | 64 | 0.23878 | 0.67813 | 0.48443 | 0.10258 | 0.24082 | 0.18349 |
| Enc. + Dec. | 64 / 48 | 0.25178 | **0.69114** | **0.50447** | **0.11092** | **0.24312** | **0.18783** |
| Enc. + Dec. | 64 / 48 / 32 | 0.24670 | 0.68019 | 0.48613 | 0.10136 | 0.22797 | 0.17482 |
| Enc. + Dec. | 64 / 48 / 32 / 24 | 0.24224 | 0.68162 | 0.48716 | 0.10134 | 0.23662 | 0.17905 |
| Enc. + Dec. | 96 / 64 | 0.24377 | 0.68615 | 0.49529 | 0.09347 | 0.22416 | 0.16551 |
| Enc. + Dec. | 96 / 64 / 48 | 0.24308 | 0.68265 | 0.48681 | 0.10228 | 0.22966 | 0.17186 |
| Enc. + Dec. | 128 / 96 | 0.24416 | 0.68858 | 0.50031 | 0.09559 | 0.22862 | 0.17767 |

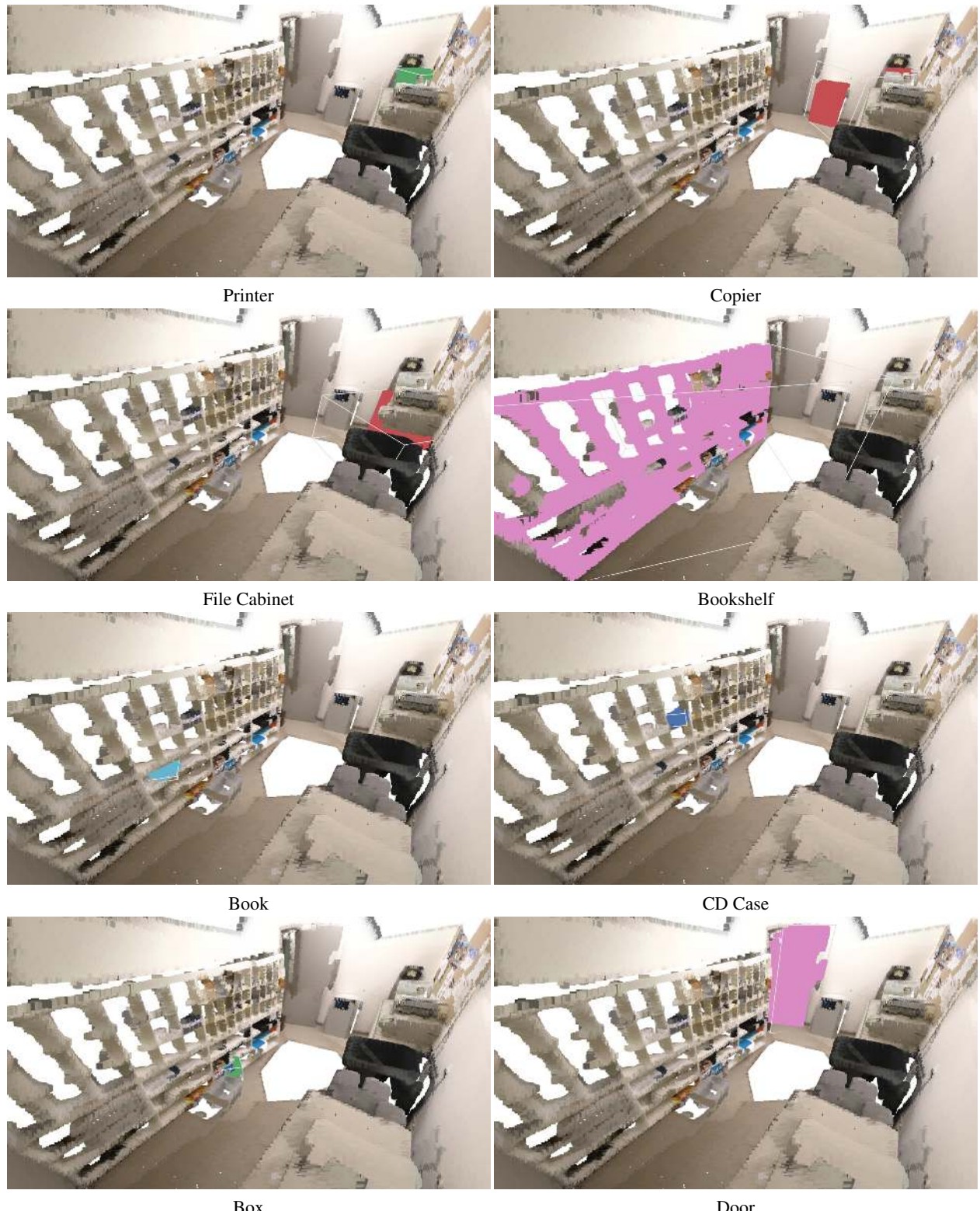

Printer

Copier

File Cabinet

Bookshelf

Book

CD Case

Box

Door

*Figure 16.* **Additional qualitative results (Scene 0462, Copy Room).** Office equipment and furniture.

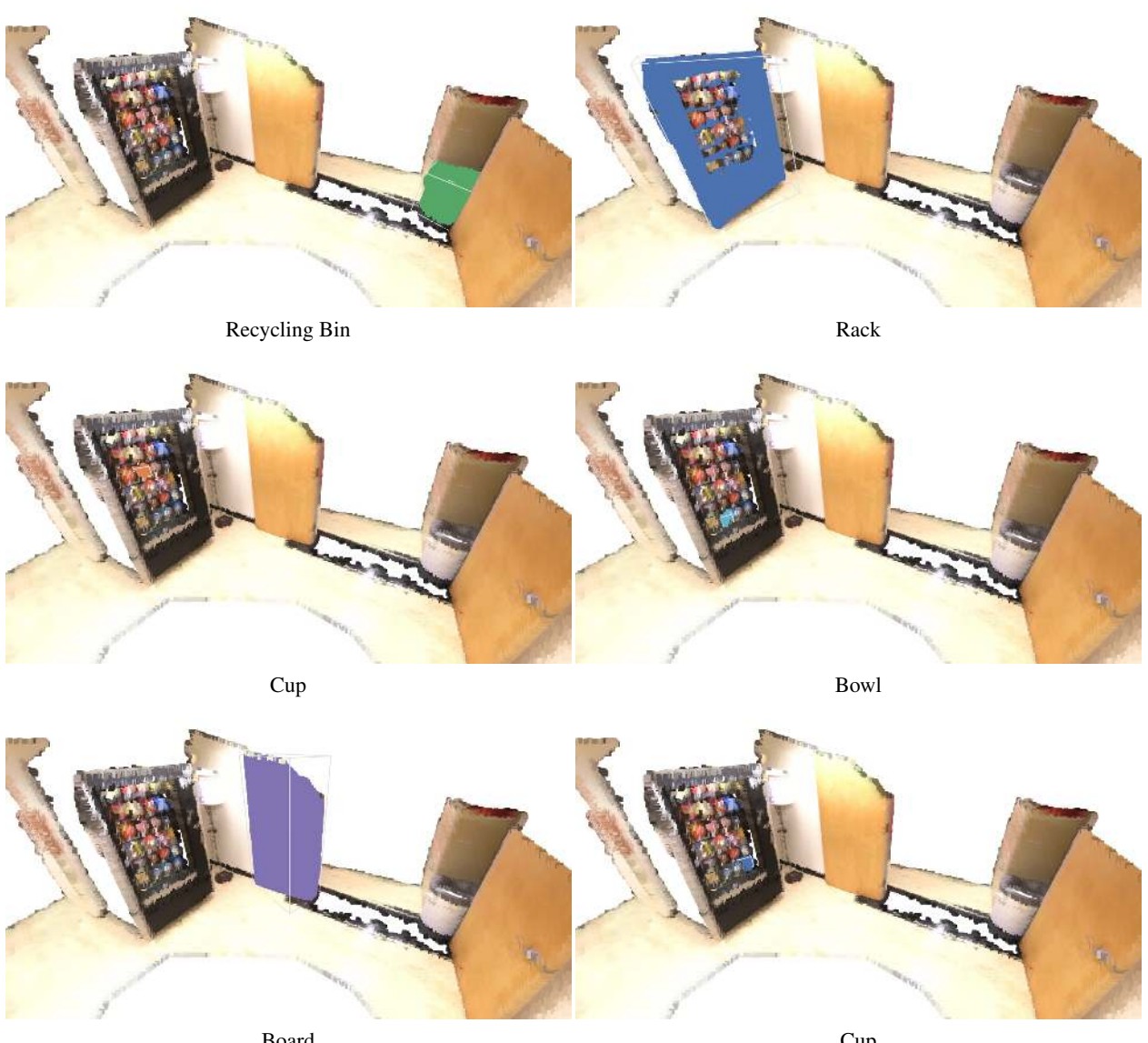

*Figure 17.* **Additional qualitative results (Scene 0019, Kitchenette).** Small objects: cups, bowls.

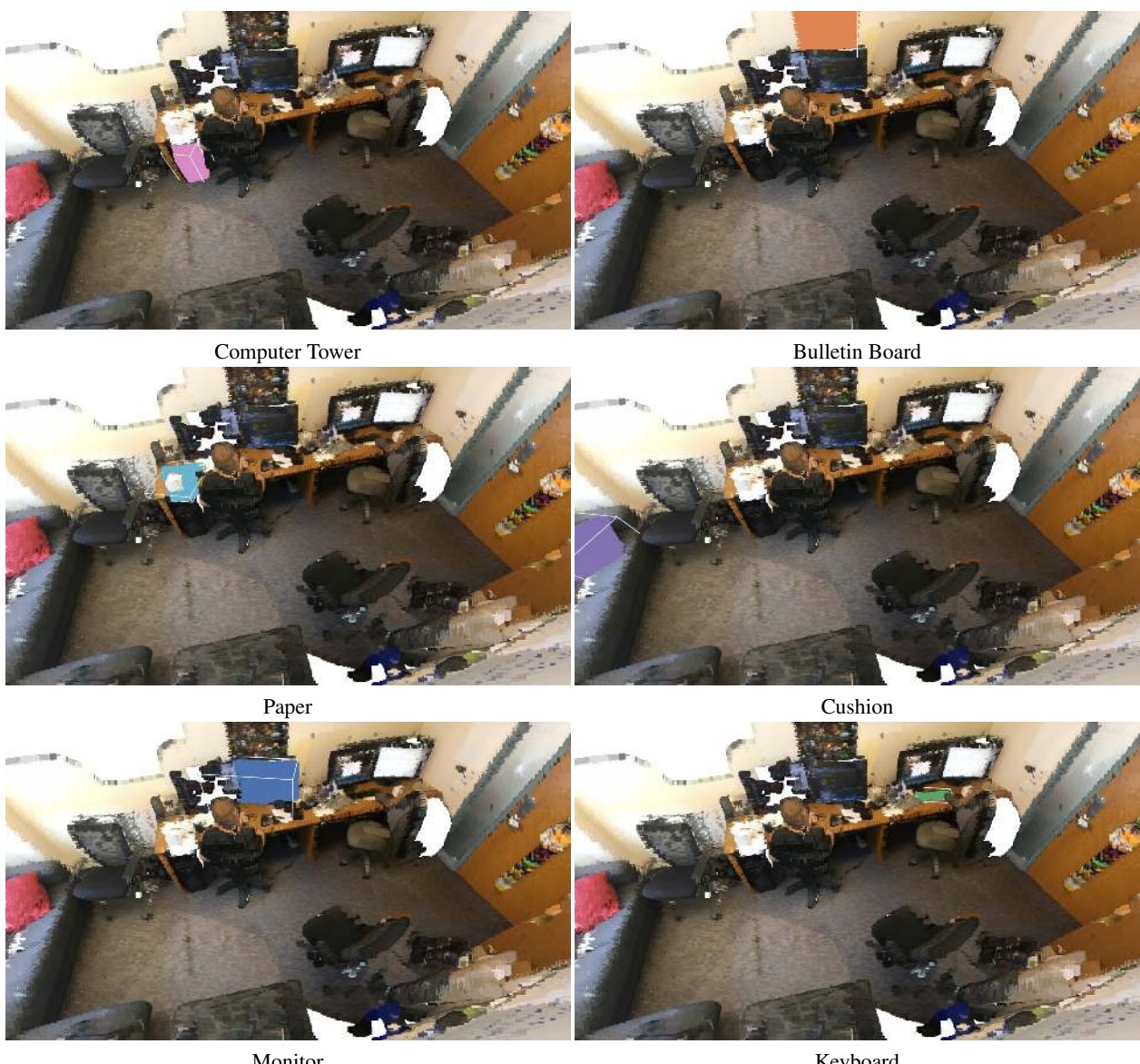

*Figure 18.* **Additional qualitative results (Scene 0474, Office).** Computer peripherals and furniture.

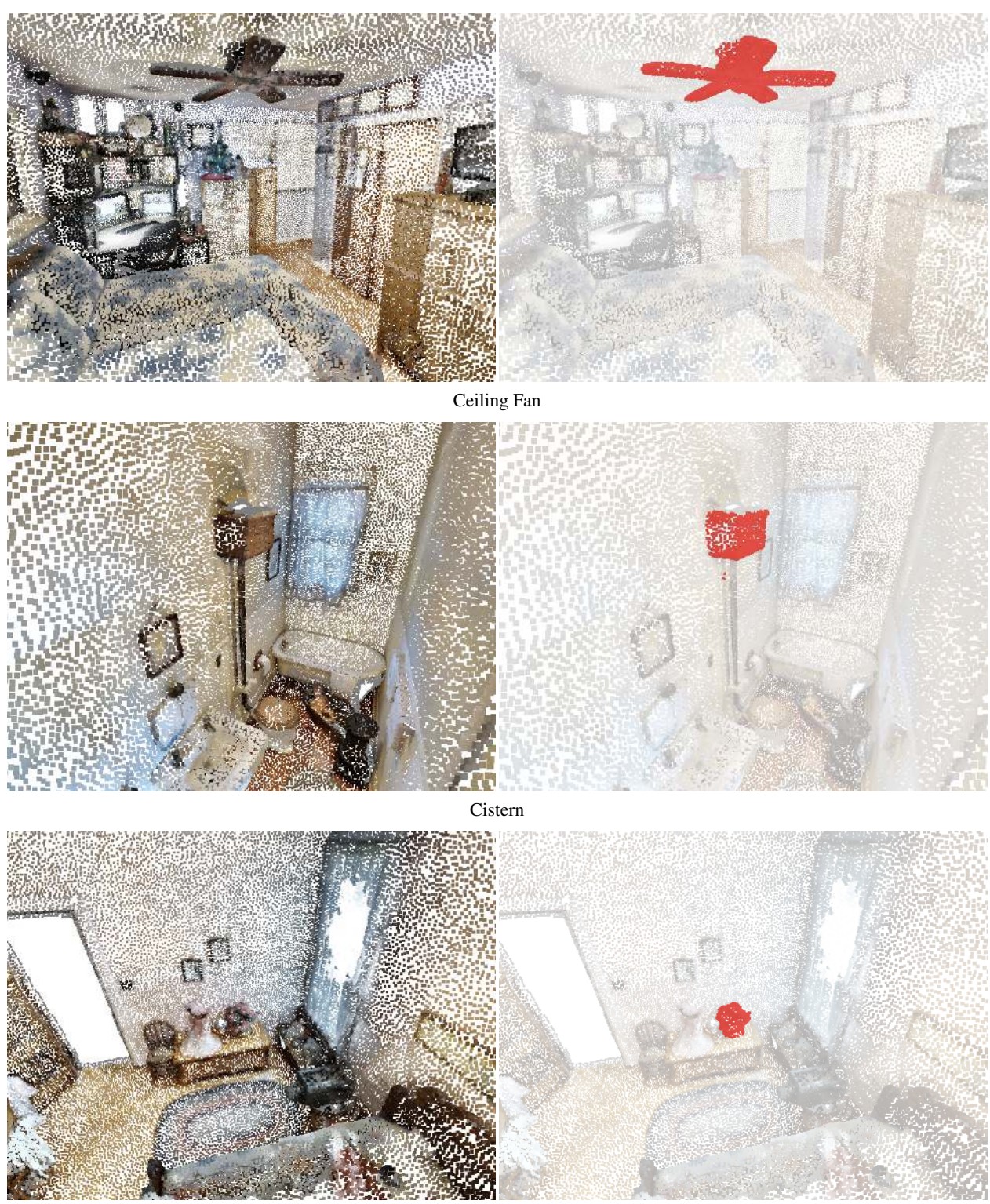

Ceiling Fan

Cistern

Flower

*Figure 19.* **Additional novel category segmentation on Matterport3D (1/2).** Layout follows Fig. 15.

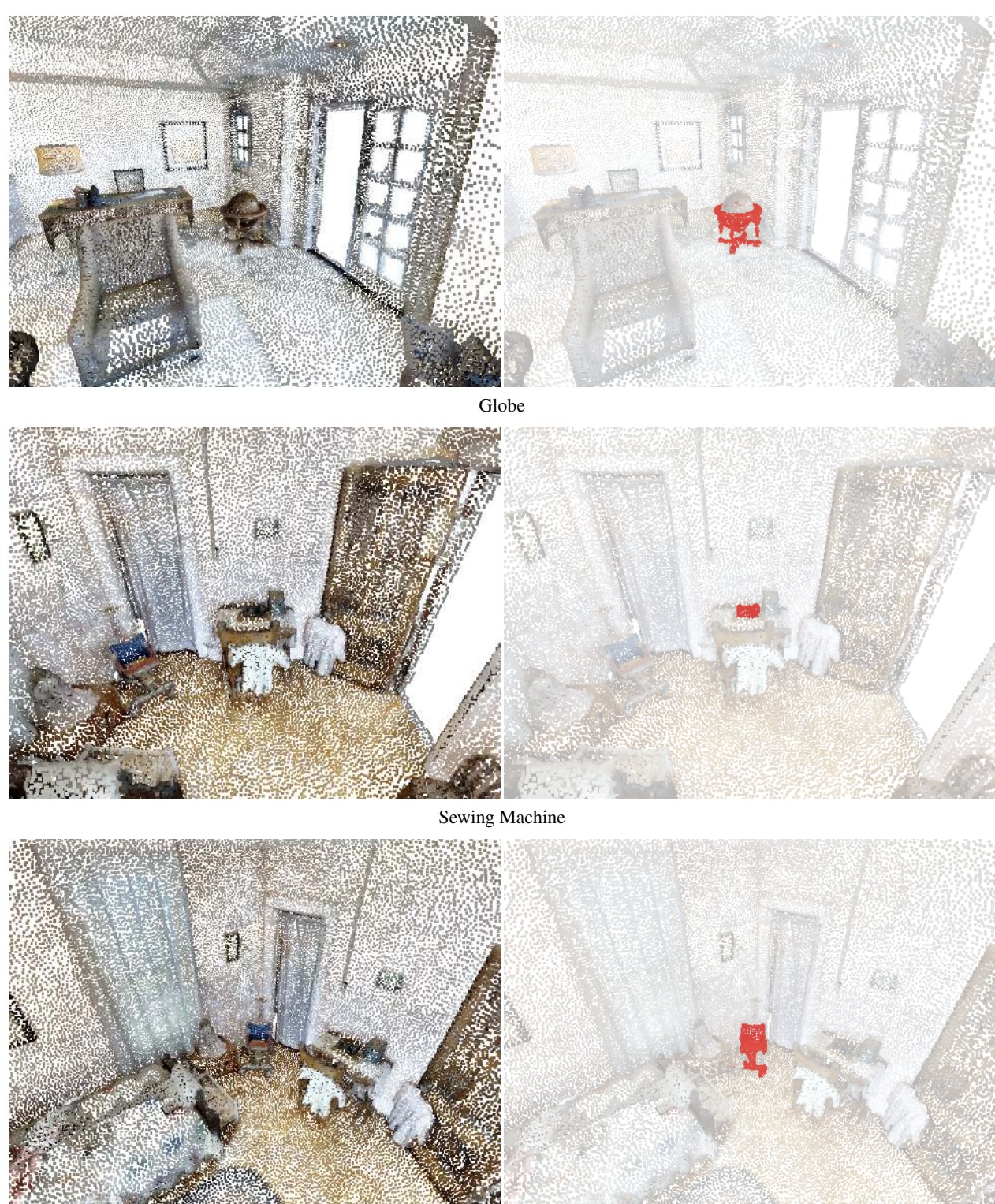

Globe

Sewing Machine

Stroller

*Figure 20.* **Additional novel category segmentation on Matterport3D (2/2).** Layout follows Fig. 15.

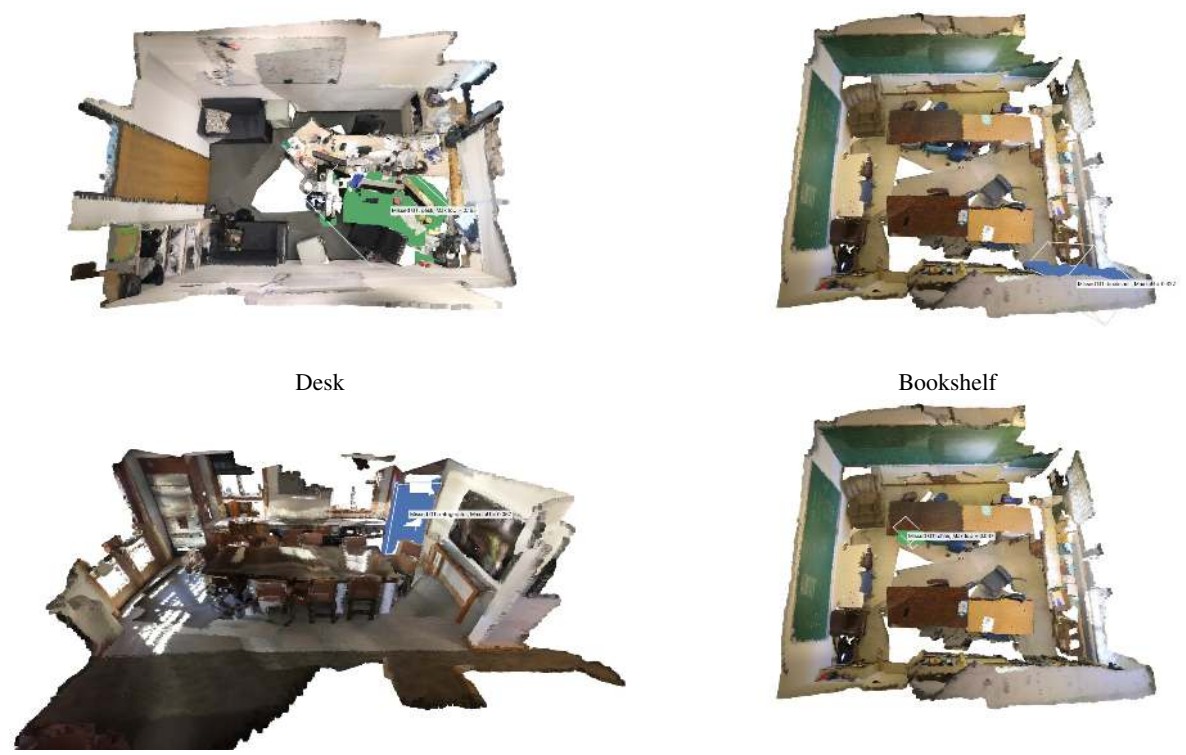

Desk    Bookshelf

Refrigerator    Chair

*Figure 21.* **Missed ground truth visualizations.** We visualize examples where the network predictions failed to achieve an IoU > 0.5. These cases include challenging instances such as partially occluded desks or geometrically complex bookshelves.

*Table 14.* Validation mIoU (%) on ScanNet20, ScanNet++, ScanNet200, and Matterport3D.

| Method | ScanNet20 | ScanNet++ | ScanNet200 | Matterport3D |
|---|---|---|---|---|
| PTv3 | 63.71 | 24.52 | 14.77 | 13.17 |
| Space-Curve Transformer | **65.46** | **25.84** | **16.55** | **13.95** |
| Space-Curve Transformer + 256C | **65.46** | 25.84 | 16.55 | 13.95 |
| Space-Curve Transformer + 512C | 63.28 | 25.68 | 16.9 | 13.74 |
| Space-Curve Transformer + 768C | 64.02 | **27.00** | 17.91 | **15.13** |
| Space-Curve Transformer + 1024C | 65.19 | 26.07 | **18.12** | **15.13** |
| Space-Curve Transformer 512C + 16 | **65.68** | 25.81 | 16.57 | 13.87 |
| Space-Curve Transformer 512C + 32 | 63.28 | 25.68 | 16.9 | 13.74 |
| Space-Curve Transformer 512C + 64 | 63.67 | 26.42 | 17.49 | **14.09** |
| Space-Curve Transformer 512C + 64 + 32 | 64.78 | **26.92** | **17.52** | 13.36 |
| Space-Curve Transformer 512C + 64 + 64 | 65.08 | 26.10 | 17.25 | 13.52 |
| Space-Curve Transformer 768C + 32 | 65.11 | 26.74 | 18.17 | 14.2 |
| Space-Curve Transformer 768C + 64 | **67.3** | 28.13 | 19.04 | 13.93 |
| Space-Curve Transformer 768C + 64,32 | 65.71 | 27.61 | 18.62 | 14.06 |
| Space-Curve Transformer 768C + 64,48,32 | 65.14 | **29.13** | 18.82 | 14.27 |
| Space-Curve Transformer 768C + 64,48,32,24 | 65.24 | 27.34 | 18.41 | 13.89 |
| Space-Curve Transformer 768C + 64,48,32,24,16 | 66.4 | 28.15 | 18.59 | **14.76** |
| Space-Curve Transformer 768C + 96 | 66.71 | 27.47 | **19.47** | 13.77 |
| Space-Curve Transformer 1024C + 64 | 65.98 | **28.56** | 19.52 | 14.87 |
| Space-Curve Transformer 1024C + 96 | 65.02 | 28.05 | **19.94** | 14.53 |
| Space-Curve Transformer 1024C + 128 | **66.32** | 28.37 | 19.9 | **15.18** |

*Table 15.* **Class-agnostic instance segmentation on ScanNet++ and Replica.** AP evaluates mask quality independent of semantic labeling. ScanNet++ baselines from MaskClustering (Yan et al., 2024), Any3DIS (Lu et al., 2025), SAI3D (Yin et al., 2024), and SAM2Object (Jia et al., 2025). * uses closed-vocabulary Mask3D proposals trained with GT.

| Method | Input | ScanNet++ (100 classes) | | | | Replica (8 scenes) | | | |
|---|---|---|---|---|---|---|---|---|---|
| | | AP | $AP_{50}$ | $AP_{25}$ | Recall | AP | $AP_{50}$ | $AP_{25}$ | Recall |
| SAI3D (2024) | 3D+2D | 17.1 | 31.1 | 49.5 | – | – | – | – | – |
| OVIR-3D (2023) | 3D+2D | 19.4 | 34.1 | 46.5 | – | – | – | – | – |
| SAM2Object (2025) | 3D+2D | 20.2 | 34.1 | 48.7 | – | – | – | – | – |
| Any3DIS (2025) | 3D+2D | 22.2 | 35.8 | 47.0 | 40.8 | – | – | – | – |
| OpenMask3D* (2023) | 3D+2D | 22.8 | 33.3 | 45.7 | – | – | – | – | – |
| MaskClustering* (2024) | 3D+2D | 27.9 | 42.8 | 54.7 | – | – | – | – | – |
| SpaCeFormer (Ours) | 3D only | **29.8** | **54.8** | **75.1** | **74.7** | **33.2** | **56.1** | **68.7** | **69.8** |

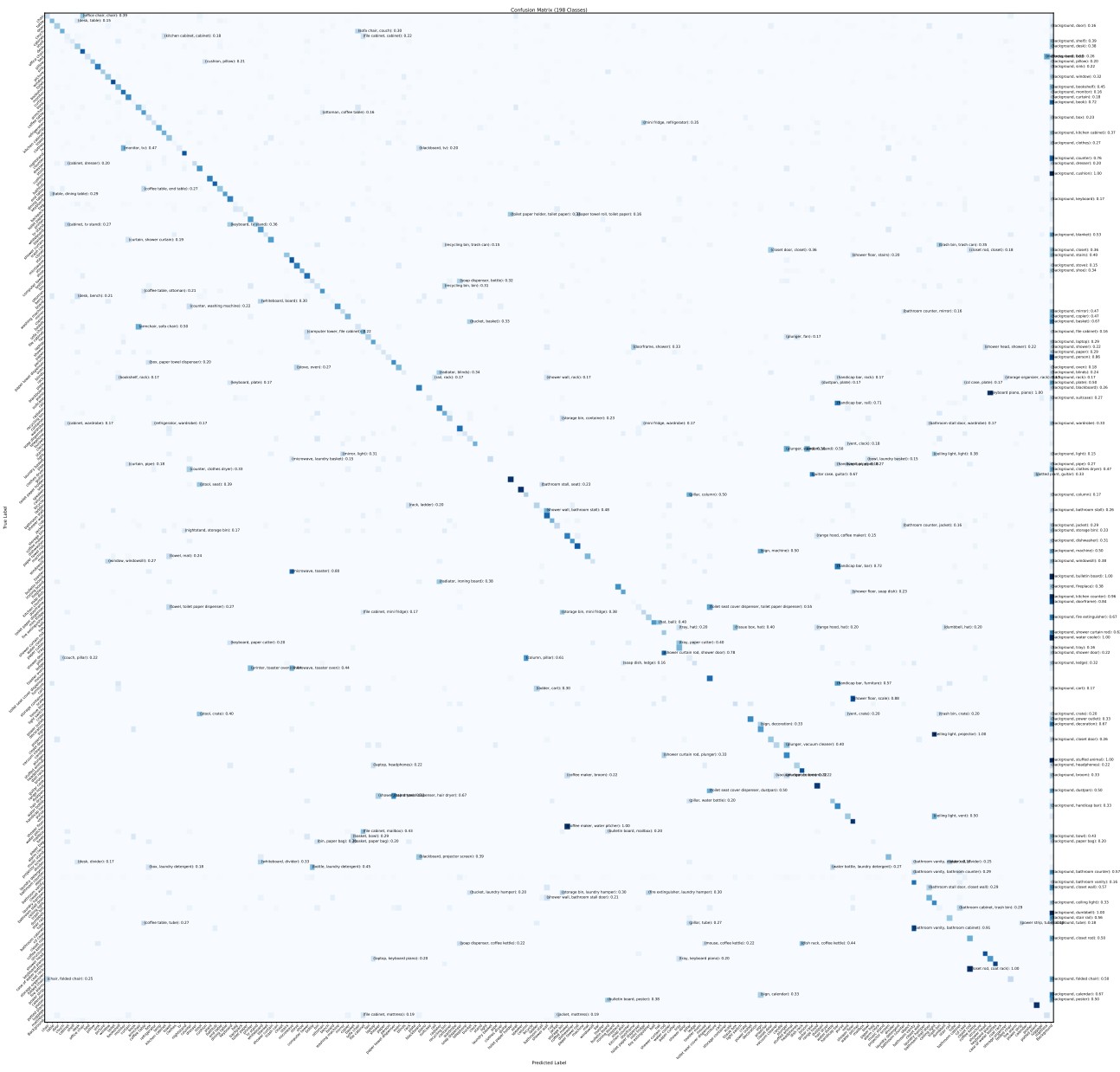

*Figure 22.* Confusion matrix of the validation set predictions. The values are row-normalized (recall). The model achieves high accuracy on distinct large objects (e.g., bed, wall, floor) but shows some confusion between geometrically similar categories.

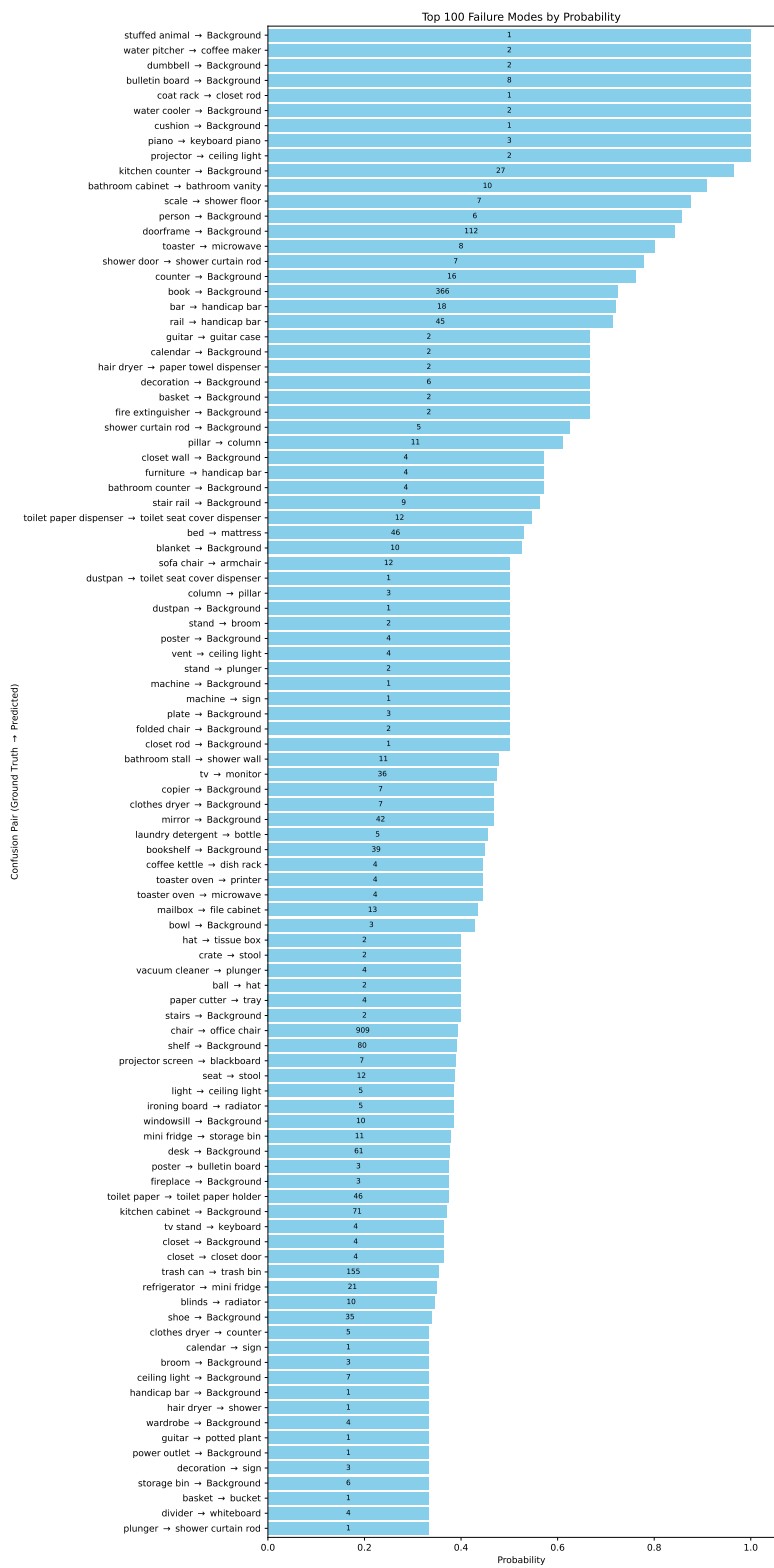

*Figure 23.* Top 100 most confused prediction pairs. The bar length represents the confusion probability, while the number inside each bar indicates the absolute count of failure cases. The y-axis labels show the Ground Truth → Prediction class assignments.

