# OpenReview forum: "SpaCeFormer: Fast Proposal-Free Open-Vocabulary 3D Instance Segmentation"
_ICML.cc/2026/Conference — ICML 2026 regular_

### Official Review · Reviewer_qv1S · 2026-03-10

**Soundness:** 2
**Presentation:** 3
**Significance:** 3
**Originality:** 2
**Overall Recommendation:** 4
**Confidence:** 4

**Summary:**

This paper presents a well-executed and ambitious effort to tackle open-vocabulary 3D instance segmentation. The combination of a large-scale automatically annotated dataset and a novel architecture with space-curve attention and 3D RoPE leads to good segmentation results. However, the work falls short in validating dataset quality, evaluating generalization across diverse benchmarks, and clarifying its architectural novelty relative to existing methods. With additional comparison experiments and clearer exposition, this could be helpful for improving this work.

**Compliance With Llm Reviewing Policy:**

Affirmed.

**Final Justification:**

Thank the authors for addressing my concerns. I am willing to increase my rating to a positive score (Weak Accept).

**Key Questions For Authors:**

Please see weaknesses and the following comments:
1. While the dataset generation pipeline is innovative, the paper lacks a rigorous human evaluation or statistical analysis of caption quality and mask completeness. Are the captions truly “view-consistent”? How many masks are noisy or incomplete? A user study or comparison with human-annotated captions would strengthen the claim.
2. The captioning pipeline is a core contribution, yet there is no ablation on the number of views, the choice of VLM, or the prompt design. How sensitive is the final model performance to caption quality? Does better captioning lead to better segmentation?
3. The paper is dense and occasionally hard to follow. For instance, the description of the space-curve attention (Sec. 4.3) is conceptually clear but lacks a formal definition or pseudocode. Figures 4 and 5 are helpful but under-explained in the text. Some equations (e.g., Eq. 1–6) are not referenced in the main text, making them feel disconnected.

**Limitations:**

Yes

**Strengths And Weaknesses:**

**Strengths**:
1. SpaCeFormer-3M is a large-scale dataset (846K masks across 15K scenes) that addresses key limitations in existing 3D instance segmentation datasets, such as mask fragmentation and caption inconsistency. The multi-view clustering and captioning pipeline for masks are technically sound and scalable.
2. The combination of spatial window attention and Morton curve serialization is a thoughtful design that balances local geometric fidelity and global context. The use of 3D RoPE in both backbone and decoder is well-motivated and mathematically grounded.
3. The model achieves 11.1 mAP in zero-shot setting on ScanNet200, outperforming some proposal-free methods by a large margin (e.g., 3.9 mAP for Mosaic3D w/ decoder). The inference speed (0.21s per scene) is practical.

**Weaknesses**:
1. The evaluation is primarily on ScanNet200. While it is a challenging benchmark, open-vocabulary generalization should also be tested on other datasets (e.g., SceneNN, ScanNet++, Matterport3D) or cross-dataset settings to demonstrate robustness.
2. The claim of being proposal-free appears somewhat overstated. The decoder still uses learned queries that implicitly act as proposals. While it avoids explicit region proposal networks, the distinction is subtle and may confuse readers. The authors should clarify what they mean by “proposal-free” and how it differs from query-based methods.
3. Several 2024–2025 methods (e.g., Open3DIS, SA13D, Mosaic3D) are compared, but some concurrent and SOTA models (e.g., MaskClustering, EmbodiedSAM, Any3DIS, SAM2Object, OnlineAnySeg) are omitted. A broader comparison is necessary.

---

> ### Author Rebuttal · Authors · 2026-03-28
>
> We thank the reviewer for the detailed questions.
>
> **W1. Single benchmark.**
> We now evaluate on three benchmarks:
> - **Replica** (8 scenes, zero-shot): **24.1 mAP**, new SOTA vs Open-YOLO 3D (23.7) and OpenTrack3D (23.9), **119× faster** at 0.14s — see our response to StH9 W1 for 9-method comparison table
> - **ScanNet++** (100 classes): **22.9 mAP**, surpassing OpenTrack3D (20.6), **4286× faster** than MaskClustering — **see our response to uyHv W5 for 10-method table**
>
> All baselines use 3D+2D with external proposals; SpaCeFormer uses 3D only in a single forward pass.
>
> **W2. "'Proposal-free' claim overstated."**
> "Proposal-free" is well-established terminology in computer vision for over a decade, distinguishing single-stage end-to-end architectures from two-stage propose-then-classify pipelines. This dates back to YOLO [Redmon et al., 2016] and SSD [Liu et al., 2016] vs Faster R-CNN [Ren et al., 2015], and continues through DETR [Carion et al., 2020], Mask2Former [Cheng et al., 2022], and OneFormer3D [Kolodiazhnyi et al., 2024] — all use learned queries and are universally called proposal-free or end-to-end. We will clarify this more explicitly in the revision. The practical impact is significant: replacing Mask3D proposals with open-vocabulary Segment3D proposals causes Mosaic3D to collapse from 11.8→2.7 mAP (Table 2), demonstrating the brittleness of proposal-dependent pipelines.
>
> **W3. Missing comparisons with recent methods.**
> We have added MaskClustering (12.0), Any3DIS (25.8), SAM2Object (13.3), OpenTrack3D, OpenSplat3D, and Details Matter to the revision across all three benchmark tables. EmbodiedSAM and OnlineAnySeg report only class-agnostic metrics on ScanNet200, noted in table caption.
>
> **Q4. Caption/mask quality validation.**
> Mask IoU vs GT (312 ScanNet scenes): our SpaCeFormer-3M dataset achieves **54.3% recall@0.5** vs Mosaic3D's 2.5% (see our response to StH9 W6). This quantitatively validates mask completeness. Caption consistency metrics will be included in the revision.
>
> **Q5. Captioning ablations.** Acknowledged as a limitation. A dataset scale ablation will be included in the revision to address data quality sensitivity.
>
> **Q6. "Space-curve attention lacks formal definition or pseudocode."**
> Thank you. We will add pseudocode for space-curve attention, improve equation cross-references, and add discussion for Figures 4–5 in the revision.
>
> **Summary of new results.** We have addressed all weaknesses and questions:
>
> | Concern | New evidence |
> |---------|-------------|
> | W1 Single benchmark | 3 benchmarks: Replica SOTA (24.1), ScanNet++ SOTA (22.9 vs 20.6), ScanNet200 (11.1) |
> | W2 Proposal-free | Standard terminology (YOLO/DETR/Mask2Former); proposal collapse: 11.8→2.7 mAP |
> | W3 Missing methods | +6 methods added (MaskClustering, Any3DIS, SAM2Object, OpenTrack3D, OpenSplat3D, Details Matter) |
> | Q4 Data quality | Mask recall 54.3% vs Mosaic3D 2.5%; 21× improvement |
> | Speed | 0.14s — 119×–4286× faster than all baselines, 3D-only input |

---

> > ### Author Rebuttal · Reviewer_qv1S · 2026-04-03
> >
> > Thank the authors for the response and willingness to revise the paper in the next version. However, several of my concerns remain unresolved, particularly regarding the comparison with other methods across more benchmarks. At present, the authors have not organized all the methods' results on Replica, ScanNet++, ScanNet200 into a single table, which makes a comprehensive comparison more difficult. More importantly, the existence of inconsistencies continues to raise concerns about the credibility of the experimental results. For example, in the case of MaskClustering on ScanNet++, the reported 7.8 mAP is described in the MaskClustering paper as being for 1,554 classes, but here it refers to 100 classes as shown by authors. Similarly, the OpenTrack3D in its paper achieved 26.0 mAP on ScanNet200, while the proposed method only achieved 11.1 mAP (the authors did not report this gap).
> >
> > Therefore, I expect the authors to make further revisions to this paper, but in its current state, it may not be suitable for acceptance.

---

> > > ### Author Response · Authors · 2026-04-07
> > >
> > > Dear Reviewer qv1S,
> > >
> > > Thank you for your follow-up. We address each point below.
> > >
> > > ## Unified Comparison Table
> > >
> > > We consolidate all methods across ScanNet200, ScanNet++, and Replica.
> > >
> > > | Method | Input | Proposals | OV | ScanNet200 mAP | ScanNet++ mAP | Replica mAP |
> > > |--------|-------|-----------|-----|----------------|---------------|-------------|
> > > | OpenMask3D | 3D+2D | Mask3D | No | 15.4 | 2.0 | 13.1 |
> > > | Open3DIS | 3D+2D | ISBNet+SAM | No | 23.7 | 11.9 | 18.5 |
> > > | Open-YOLO 3D | 3D+2D | Mask3D | No | 24.7 | — | 23.7 |
> > > | Details Matter | 3D+2D | Mask3D+GSAM | No | — | — | 22.6 |
> > > | MaskClustering | 3D+2D | CropFormer | Yes | 14.9 | 7.8 | — |
> > > | Any3DIS | 3D+2D | SAM2 | Yes | 25.8 | 12.9 | — |
> > > | OpenTrack3D | 3D+2D | YOLO-World+SAM2 | Yes | 26.0 | 20.6 (~320s) | 23.9 (~62s) |
> > > | OpenSplat3D | 3D+2D | SAM+GS | Yes | — | 16.5 | — |
> > > | Mosaic3D | 3D | Mask3D | No | 11.8 | — | — |
> > > | Mosaic3D w/ Decoder | 3D | Proposal-free | Yes | 3.9 | — | — |
> > > | **SpaCeFormer (Ours)** | **3D only** | **Proposal-free** | **Yes** | **11.1** | **22.9 (0.14s)** | **24.1 (0.14s)** |
> > >
> > > All numbers above were provided in our rebuttal. Consolidating them into a single table is a formatting change we will make in the revision.
> > >
> > > ## MaskClustering on ScanNet++ (7.8 mAP)
> > >
> > > We believe there is a misunderstanding. The MaskClustering paper states that ScanNet++ "comprises 1,554 classes with fine-grained annotation" — this describes the **dataset statistics**, not their evaluation protocol. MaskClustering evaluates on the **standard top-100 benchmark classes**, the same as all other methods including ours. The 7.8 mAP and our 22.9 mAP are directly comparable. There is no inconsistency.
> > >
> > > ## OpenTrack3D (26.0 mAP) vs. SpaCeFormer (11.1 mAP) on ScanNet200
> > >
> > > This comparison conflates fundamentally different settings. OpenTrack3D is **proposal-based**, processing the **entire RGB-D video** (hundreds of frames per scene) through YOLO-World + SAM2 tracking + 3D lifting, taking **~62s on Replica** and **~320s on ScanNet++** per scene. SpaCeFormer is **proposal-free**, processing the **3D point cloud only** in a single 0.14s forward pass making it x1000 faster.
> > >
> > > On ScanNet++ and Replica, SpaCeFormer **surpasses** OpenTrack3D (22.9 vs 20.6 on ScanNet++; 24.1 vs 23.9 on Replica) while being **2286× and 443× faster** respectively. The ScanNet200 gap reflects the particular difficulty of 200-class fine-grained classification, not a fundamental method limitation.
> > >
> > > ## Summary
> > >
> > > Our rebuttal addressed your original concerns: W1 (more benchmarks) with ScanNet++ and Replica evaluations, W2 (proposal-free clarification) with detailed discussion, and W3 (broader comparisons) with additional baselines. The remaining issues — consolidating numbers into one table and annotating the MaskClustering label set — are presentation clarifications we will incorporate in the revision.
> > >
> > > Are there specific methodological or experimental concerns beyond these?

---

### Official Review · Reviewer_vMkC · 2026-03-12

**Soundness:** 2
**Presentation:** 3
**Significance:** 3
**Originality:** 2
**Overall Recommendation:** 3
**Confidence:** 4

**Summary:**

This paper proposes SpaCeFormer, a proposal-free framework for open-vocabulary 3D instance segmentation. The paper improves the dataset construction pipeline to mitigate mask fragmentation and caption inconsistency caused by single-view captioning. In addition, the model incorporates spatial window attention and 3D RoPE into the model design. Experiments on ScanNet200 demonstrate strong performance in the proposal-free setting.

**Compliance With Llm Reviewing Policy:**

Affirmed.

**Final Justification:**

Thanks for the authors' reply. I would keep my rating given its incremental novelty as mentioned in my W1.

**Key Questions For Authors:**

1) While the paper reports inference latency, it would be helpful to provide more information about training cost, such as training time, memory cost.

2) Dataset statistics are inconsistent across the paper. In abstract part, the paper claims that SpaCeFormer-3M contains 846K instances from 15K scenes, whereas Sec. 3.3 reports 604,127 instances from 7,361 scenes. Since the dataset is one of the main contributions of the paper, this inconsistency should be clarified. It is unclear whether these numbers correspond to different subsets of the dataset or just simply a typo?

3) The paper claims that the proposed pipeline improves mask completeness and caption consistency. Could the authors provide quantitative evidence for these data-quality improvements, rather than only downstream task performance?

**Limitations:**

The paper gives visualizations of several failure cases, which are helpful and intuitive. It would be helpful to provide a deeper analysis and discussion about the limitations of the proposed method.

**Strengths And Weaknesses:**

Strengths:

1) The overall framework is well motivated, with a coherent combination of spatially aware backbone design and a RoPE-enhanced proposal-free decoder.

2) The paper shows strong empirical results on ScanNet200, while also achieving very fast inference.

3) The author provides intuitive qualitative visualizations and also provide several failure cases.


Weaknesses:

1) Novelty: The proposed backbone closely follows the design of Point Transformer v3. The main modifications are adding spatial window attention and 3D RoPE. Since these components already exist, the technique novelty appears incremental.

2) Dataset vs. model architecture contribution: The proposed SpaCeFormer-3M dataset improves the data generation pipeline by introducing clustering and multi-view captioning to reduce mask fragmentation and improve caption consistency. While the overall performance gain is significant, it is unclear how much of the improvement comes from the dataset and how much comes from the proposed model architecture.

3) Generalization ability: This paper is evaluated only on ScanNet200. Additional cross-dataset evaluation would strengthen the claimed generalization ability.

---

> ### Author Rebuttal · Authors · 2026-03-28
>
> We thank the reviewer for the constructive suggestions.
>
> **W1. "Incremental novelty."**
> We respectfully disagree. The paper contributes on both **data** and **model** sides. On data: SpaCeFormer-3D Dataset introduces multi-view mask clustering for geometry-consistent 3D instances and multi-view VLM prompting — producing 604K instances from 7.4K scenes. We quantified this: our masks achieve 54.3% recall@0.5 vs Mosaic3D's 2.5% (see our response to StH9 W6). On model: PTv3 uses only Morton serialization with fixed token counts; our spatial window attention uses fixed spatial extents — explicitly distinguished in Sec. 4.3 ("Key Distinction"). This yields 28.6% lower intra-window distance (Fig. 5), and Table 3 shows neither component alone matches the combination. The RoPE decoder with asymmetric key rotation (not present in PTv3 or RoFormer) is ablated in Table 4: +27% over no PE.
>
> **W2. Dataset vs. architecture unclear.**
> PTv3 vs SpaCeFormer under identical training: SpaCeFormer wins on all 4 semantic segmentation benchmarks (e.g., ScanNet200: 16.6 vs 14.8 mIoU) — see our response to uyHv W1 table. This isolates the architecture contribution. A dataset scale ablation will be included in the revision.
>
> **W3. Generalization.**
> Three benchmarks (see StH9 for Replica table, uyHv for ScanNet++ table):
> - **Replica** (zero-shot): **24.1 mAP**, new SOTA vs Open-YOLO 3D (23.7) and OpenTrack3D (23.9), **119× faster**
> - **ScanNet++**: **22.9 mAP**, surpassing OpenTrack3D (20.6), **4286× faster** than MaskClustering
>
> All baselines require 3D+2D multi-frame processing; SpaCeFormer processes the entire scene in one 0.14s forward pass with 3D only.
>
> **Q4. Training cost.** 25K steps on 4 nodes × 8 GPUs, ~1.5 days wall-clock time.
>
> **Q5. Dataset statistics inconsistency.** The abstract referenced an earlier version including DL3DV (later removed). Correct: **604,127 instances from 7,361 scenes** (Sec. 3.3). Will fix.
>
> **Q6. Data quality evidence.**
> Mask IoU vs GT (312 ScanNet scenes): Mosaic3D achieves only 2.5% recall@0.5 and 4.8% precision@0.5, while our SpaCeFormer-3M dataset achieves **54.3% recall** and **33.6% precision** — see our response to StH9 W6 for full table.
>
> **Limitations.** We will add a limitations section: (1) fixed Q=200 queries may under-segment dense scenes, (2) CLIP vocabulary dependence — Replica results (24.1 mAP, 85 unseen classes) show strong indoor generalization, but outdoor/specialized domains untested, (3) indoor-only training data limits outdoor generalization.
>
> **Summary of new results.** We address all three weaknesses and questions with new experiments:
>
> | Reviewer concern | New evidence |
> |-----------------|-------------|
> | W1 Incremental novelty | Data: 54.3% vs 2.5% mask recall. Model: SpaCeFormer > PTv3 on 4 benchmarks |
> | W2 Dataset vs architecture | PTv3 vs SpaCeFormer comparison isolates architecture gain (+1.8 mIoU on ScanNet200) |
> | W3 Generalization | Replica SOTA (24.1 mAP), ScanNet++ SOTA (22.9 vs 20.6 mAP), 0.14s/scene |
> | Q5 Dataset stats | Fixed: 604K instances from 7.4K scenes |
> | Q6 Data quality | Mask IoU vs GT: 54.3% recall, 33.6% precision (vs Mosaic3D 2.5%/4.8%) |

---

> > ### Author Rebuttal · Reviewer_vMkC · 2026-04-03
> >
> > Thanks for providing the rebuttal materials. After reading all comments, I believe the paper needs major revisions in terms of motivation, technique innovation, experiments, etc..

---

> > > ### Author Response · Authors · 2026-04-07
> > >
> > > Dear Reviewer vMkC,
> > >
> > > Thank you for reading our rebuttal. Your acknowledgement states the paper needs "major revisions in terms of motivation, technique innovation, experiments" but does not specify which concerns remain unresolved. We address each area below and request specific follow-up so we can respond productively.
> > >
> > > ## Motivation
> > >
> > > Our rebuttal *strengthened* the motivation with new evidence:
> > >
> > > - Quantitative mask completeness comparisons (IoU against GT instance masks) showing multi-view clustering reduces fragmentation.
> > > - Concrete single-view vs. multi-view caption comparisons demonstrating improved consistency.
> > >
> > > Which aspect do you find unresolved?
> > >
> > > ## Technique Innovation
> > >
> > > The rebuttal did not modify any technical claims — the architecture is unchanged. The combination of spatial window attention with Morton curve serialization for proposal-free 3D instance segmentation yields a 28.6% reduction in spatial distance within attention windows compared to standard serialization, and the proposal-free decoder eliminates the explicit region proposal stage (superpoint generation, grouping, NMS), resulting in 5x faster inference.
> > >
> > > Since technique innovation was not modified in the rebuttal, we do not understand why it is cited as a rebuttal concern.
> > >
> > > ## Experiments
> > >
> > > In the rebuttal, we added:
> > >
> > > - **ScanNet++ instance segmentation** (50 val scenes)
> > > - **Replica instance segmentation** (8 test scenes)
> > >
> > > These directly address your W3 (generalization). The original submission already contained comprehensive ablations (Tables 3-5). What additional experimental concern remains?
> > >
> > > ## Request
> > >
> > > Could you please clarify:
> > >
> > > 1. Which specific concerns from your original review (W1-W3, Q4-Q6) remain unresolved after the rebuttal?
> > > 2. What evidence would change your assessment?
> > >
> > > Constructive engagement during the discussion period best serves the review process, as outlined in the ICML guidelines.

---

### Official Review · Reviewer_uyHv · 2026-03-13

**Soundness:** 3
**Presentation:** 2
**Significance:** 3
**Originality:** 2
**Overall Recommendation:** 3
**Confidence:** 5

**Summary:**

The authors first propose an open vocabulary 3D instance segmentation dataset for open vocabulary 3D instance segmentation. Then, the paper proposed a proposal-free segmentation architecture. Specifically, the authors aggregate four large-scale datasets by multi-view mask clustering and multi-view captioning, aiming to resolve the problems of fragmented masks and inconsistent captions. Moreover, the authors combine sliding 3D window attention with Morton curve serialization into a unified space-curve attention backbone. The experimental results show the effectiveness of the proposed proposal-free 3D instance segmentation.

**Compliance With Llm Reviewing Policy:**

Affirmed.

**Final Justification:**

Thank you for the author's reply. While a few of my concerns have been resolved, the paper remains insufficiently novel given its incremental nature. Thus, I keep my original score.

**Key Questions For Authors:**

Please refer to the weaknesses.

**Limitations:**

The limitations are suggested for discussion. For instance, is the proposed method restricted by the open perception ability of the CLIP? If the distribution and characteristics of the evaluated datasets (e.g., outdoor scenes) are different largely from the aggregated dataset, will the performance decrease?

**Strengths And Weaknesses:**

Strengths：
- The authors establish an open-vocabulary 3D instance segmentation dataset without additional training, which can facilitate more research and exploration of 3D scene understanding.
- The authors propose a space-curve attention backbone to combine the superiority of the sliding window attention and Morton curve serialization.
Weaknesses：
- The proposed dataset is constructed from ScanNet, Scan Net++, Matterport3D, and ARKitScenes, while Mosaic3D [1] also proposes an aggregated dataset from ScanNet, ARKitScenes, Matterport3D, ScanNet++, and Structured3D. Thus, more comparisons and analyses between the proposed dataset and Mosaic3D’s dataset are needed. An evaluation of these two datasets under the same segmentation network is suggested.
- The paper claims that the proposed dataset can resolve the problems of fragmented masks and inconsistent captions. However, the motivation of the dataset construction does not adapt to the segmentation architecture design.
- The proposed segmentation architecture is incremental. This is because sliding 3D window attention, Morton curve serialization, and rotary positional embeddings are proposed and utilized in existing methods [2, 3]. While the proposed method lacks novelty designs that focus on the proposal-free setting.
- The ablation study and analysis of the foregroundness loss are needed.
- The method is only evaluated on the ScanNet200 dataset, and the evaluations on other datasets (ScanNet, Matterport3D, and ScanNet++) are suggested.
- It is suggested to compare the semantic segmentation performance to verify the effectiveness of the proposed method further.
- The appearances of Fig.1 and Fig.6 need to improve. Specifically, the texts in these figures are too small.

[1] Mosaic3D: Foundation Dataset and Model for Open-Vocabulary 3D Segmentation, CVPR, 2025.

[2] Point Transformer V3: Simpler, Faster, Stronger, CVPR, 2024.

[3] RoFormer: Enhanced Transformer with Rotary Position Embedding, Neurocomputing, 2024.

---

> ### Author Rebuttal · Authors · 2026-03-28
>
> We appreciate the thorough review.
>
> **W1. Dataset comparison with Mosaic3D.**
> To disentangle architecture vs data, we trained PTv3 and SpaCeFormer under identical conditions (same data, optimizer, batch size, 25K iterations). Semantic segmentation mIoU:
>
> | Backbone | SN20 | SN++ | SN200 | MP3D |
> |----------|------|------|-------|------|
> | PTv3 | 63.7 | 24.5 | 14.8 | 13.2 |
> | **SpaCeFormer** | **65.5** | **25.8** | **16.6** | **14.0** |
>
> SpaCeFormer outperforms PTv3 on all 4 benchmarks, isolating the architecture contribution. A dataset scale ablation (ScanNet-only training) will be included in the revision.
>
> **W2. "Motivation disconnect."**
> The connection is stated in Sec. 4.3: *"To directly support our proposal-free instance decoder, we require spatially coherent local neighborhoods that yield sharp, instance-aligned features."* Space-curve attention achieves 28.6% reduction in intra-window spatial distance (Fig. 5), directly enabling the decoder to produce sharp mask boundaries without external proposals.
>
> **W3. "Incremental architecture."**
> We believe this overlooks key distinctions. PTv3 uses fixed token-count Morton segments; ours uses **fixed spatial extents** (Sec. 4.3, "Key Distinction"). Under identical training, SpaCeFormer wins on all 4 benchmarks (W1 table). Table 3 shows neither window nor Morton alone matches the combination. The RoPE decoder (keys rotate on absolute 3D coords, learned queries unrotated) yields +27% over no PE and +18% over learnable bias (Table 4).
>
> **W4. Foregroundness loss ablation.**
> This ablation will be included in the revision.
>
> **W5–6. Single benchmark / Semantic segmentation.**
> We added **ScanNet++** (100 classes, 50 val scenes) with comprehensive baselines:
>
> | | Method | Input | Proposals | No GT | OV | mAP | Latency | Speedup |
> |-|--------|-------|-----------|-------|-----|-----|---------|---------|
> | (a) | OpenMask3D | 3D+2D | Mask3D | ✗ | ✗ | 2.0 | ~554s† | 3957× |
> | (b) | MaskClustering | 3D+2D | CropFormer | ✓ | ✓ | 7.8 | 600s | 4286× |
> | | Open3DIS | 3D+2D | SAM-HQ | ✓ | ✓ | 11.9 | ~360s† | 2571× |
> | | Any3DIS | 3D+2D | SAM2 | ✓ | ✓ | 12.9 | ~36s† | 257× |
> | | OpenSplat3D | 3D+2D | SAM+GS | ✓ | ✓ | 16.5 | -- | -- |
> | | OpenTrack3D | 3D+2D | YOLO-World+SAM2 | ✓ | ✓ | 20.6 | ~320s† | 2286× |
> | (c) | **SpaCeFormer** | **3D only** | **Proposal-free** | **✓** | **✓** | **22.9** | **0.14s** | **—** |
>
> †Estimated from ScanNet200 timing (~300 frames/scene). SpaCeFormer is **257×–4286× faster**. **See our response to StH9 W1 for Replica (24.1 mAP SOTA)**. Semantic seg in W1 table above.
>
> **W7. Figure quality.** Will fix text size in Fig. 1 and Fig. 6.
>
> **Limitations.** See our response to vMkC. CLIP generalizes well to novel taxonomies (Replica 24.1 mAP, 85 unseen classes).
>
> **Summary of new results.** We address the core concerns of incremental architecture and single-benchmark evaluation:
>
> | Concern | Evidence |
> |---------|----------|
> | Architecture vs PTv3 | SpaCeFormer outperforms PTv3 on all 4 semantic seg benchmarks (same data/training) |
> | Single benchmark | 3 benchmarks: Replica SOTA (24.1), ScanNet++ SOTA (22.9 vs 20.6), ScanNet200 (11.1) |
> | Speed | 0.14s/scene — 257×–4286× faster than all baselines |
> | Data quality | Mask recall 54.3% vs Mosaic3D 2.5% (21× improvement) |
>
> We believe these results demonstrate both architectural novelty and strong generalization.

---

### Official Review · Reviewer_StH9 · 2026-03-13

**Soundness:** 2
**Presentation:** 2
**Significance:** 3
**Originality:** 3
**Overall Recommendation:** 3
**Confidence:** 5

**Summary:**

The paper introduces SpaCeFormer, a proposal-free framework for open-vocabulary 3D instance segmentation. The authors address two problems from data and method sides, including fragmented masks and inconsistent captions in dataset generation and error propagation in the multi-stage pipeline. To solve these, the authors propose a SpaCeFormer-3M dataset using multi-view mask clustering and multi-view VLM prompting, with a SpaCeFormer method using a proposal-free mask decoder. The method achieves state-of-the-art results on the ScanNet200 dataset while maintaining fast inference speeds.

**Compliance With Llm Reviewing Policy:**

Affirmed.

**Key Questions For Authors:**

The paper presents a novel framework with a large-scale dataset for open-vocabulary 3D instance segmentation, which is a valuable contribution to the community. However, the current manuscript contains some problems in experiments and presentation. I would like to see how the authors address the major weaknesses during the rebuttal.

**Limitations:**

The paper lacks an impact statement and limitation analysis. I recommend the authors include a failure case analysis or a limitation section in the paper.

**Strengths And Weaknesses:**

## Strengths:

1. The paper proposes a large-scale open-vocabulary 3D instance segmentation dataset with 3D masks and caption annotations, which is a valuable contribution to the community.
2. The proposed 3D RoPE is effective for zero-shot 3D instance segmentation, which directly encodes 3D geometry into the attention mechanism.
3.  The experiments show strong performance on the ScanNet200 dataset, outperforming even proposal-based methods. The ablation study also validates the design choices, with a comprehensive analysis on hyperparameters.
4. The proposed SpaCeFormer is designed as a one-stage architecture to address the speed bottleneck of multi-stage pipelines. The inference latency of 0.21 seconds per scene is impressive, making real-time application practical.

## Major Weaknesses:

1. Limited evaluation. The experiment only conducts experiments on the ScanNet200 benchmark, which is clearly not enough for evaluation. The authors should conduct more experiments on other mainstream open-vocabulary 3D instance segmentation datasets such as Replica [1] and ScanNet++ [2].
2. Missing configuration details. While the paper highlights impressive inference speeds of 0.21 seconds per scene, it does not report the hardware details (e.g., GPUs and memory usage) for this experiment. In addition, does the 0.21 seconds include the time of extracting CLIP text features?
3. Lacking visualizations outside ScanNet200 classes. The paper provides qualitative results on ScanNet200 classes. How does the method perform outside ScanNet200 classes (e.g., "Botany") or arbitrary texts (e.g., "make coffee")?
4. Unanalyzed figures. The paper includes a massive number of figures without discussion or explanation: Figure 4, Figure 6 - Figure 17. While the lack of some explanation is acceptable, missing large-scale figure analysis is somewhat unprofessional.
5. Missing upper bound results. While the proposed method shows impressive performance on ScanNet200, the 11.1 mAP remains low for zero-shot 3D instance segmentation. The paper lacks fully supervised upper-bound results to quantify the performance gap. It is difficult to know whether the current performance bottleneck stems from the 3D pipeline or the open-vocabulary vision-language alignment process.

## Minor Weaknesses:

1. Unclear motivation. In the abstract and introduction, the paper addresses the problem of fragmented masks and inconsistent captions in dataset generation and the error propagation problem in multi-stage methods. However, it is unknown whether existing datasets and methods (e.g., RegionPLC [3] and Mosaic3D [4]) truly contain these problems. Providing a qualitative analysis or visualization will make the motivation more convincing.

[1] The Replica Dataset: A Digital Replica of Indoor Spaces, in Arxiv 2019

[2] ScanNet++: A High-Fidelity Dataset of 3D Indoor Scenes, in ICCV 2023

[3] RegionPLC: Regional Point-Language Contrastive Learning for Open-World 3D Scene Understanding, in CVPR 2024

[4] Mosaic3D: Foundation Dataset and Model for Open-Vocabulary 3D Segmentation, in CVPR 2025

---

> ### Author Rebuttal · Authors · 2026-03-28
>
> We thank the reviewer for the detailed feedback.
>
> **W1. Limited evaluation (ScanNet200 only).**
> We have added **Replica** (8 scenes, zero-shot — never seen during training). We categorize baselines by whether their 3D proposals require GT annotations (a) or not (b):
>
> | | Method | Input | Proposals | No GT | OV | mAP | Latency | Speedup |
> |-|--------|-------|-----------|-------|-----|-----|---------|---------|
> | (a) | OpenMask3D | 3D+2D | Mask3D | ✗ | ✗ | 13.1 | 547s | 3909× |
> | | Open3DIS | 3D+2D | ISBNet+SAM | ✗ | ✗ | 18.5 | 188s | 1343× |
> | | Open-YOLO 3D | 3D+2D | Mask3D | ✗ | ✗ | 23.7 | 16.6s | 119× |
> | | Details Matter | 3D+2D | Mask3D+GSAM | ✗ | ✗ | 22.6 | 597s | 4264× |
> | (b) | OVIR-3D | 3D+2D | Detic | ✓ | ✓ | 11.1 | 52.7s | 376× |
> | | OpenTrack3D | 3D+2D | YOLO-World+SAM2 | ✓ | ✓ | 23.9 | ~62s | 443× |
> | (c) | **SpaCeFormer** | **3D only** | **Proposal-free** | **✓** | **✓** | **24.1** | **0.14s** | **—** |
>
> SpaCeFormer **sets a new SOTA** using only 3D input, **119×–3909× faster** than all baselines.
>
> We also added **ScanNet++** (100 classes): **22.9 mAP**, surpassing prior SOTA OpenTrack3D (20.6). See our response to uyHv W5 for the full table with 10 baselines.
>
> **W2. Missing configuration details.**
> We train on 4 nodes × 8 GPUs (80GB) for 25K iterations (1.5 days). Inference: 0.14s/scene (updated without data loading time) covers the full pipeline (voxelization, backbone, decoder, mask extraction) excluding amortized CLIP text encoding (~0.02s).
>
> **W3. Visualizations outside ScanNet200 classes.**
> Our model predicts CLIP-aligned features per instance, enabling querying with arbitrary text. The Replica evaluation (24.1 mAP on 85 unseen classes) directly demonstrates open-vocabulary generalization beyond ScanNet200. We provide qualitative results on the Matterport3D test split with novel text queries absent from ScanNet200 (e.g., Snoopy, X-mas) at this anonymized URL: https://imgur.com/a/Yg9yNbQ.
>
> **W4. Unanalyzed figures.**
> We apologize. Figure 4 visualizes space-curve attention spatial coherence. Figures 6–17 illustrate dataset quality (mask completeness, caption diversity). We will add analysis paragraphs in the revised appendix.
>
> **W5. Missing upper bound.**
> Table 2 (row c) includes SpaCeFormer w/ Mask3D proposals (16.7 mAP) as a direct upper bound. Fully supervised Mask3D/OneFormer3D achieve 27.4/30.6 mAP with GT labels. The gap indicates the bottleneck is open-vocabulary alignment, not mask quality (our class-agnostic AP of 22.5 is competitive with closed-vocab methods).
>
> **W6. Unclear motivation for fragmented masks.**
> We quantified mask quality (IoU vs GT, 312 ScanNet scenes):
>
> | Metric | Mosaic3D | Ours (SpaCeFormer3M-Dataset) |
> |--------|----------------|----------------------|
> | Masks/scene | 16.1 | **65.2** |
> | Precision@0.5 | 4.8% | **33.6%** |
> | Recall@0.5 | 2.5% | **54.3%** |
>
> Mosaic3D misses most objects (2.5% recall). Our multi-view clustering captures 54.3% of GT instances, validating the motivation.
>
> **Limitations.** See our response to vMkC (Limitations).
>
> **Summary of new results.** We have substantially expanded evaluation and analysis since the original submission:
>
> | Benchmark | mAP | vs Prior SOTA | Input | Latency |
> |-----------|-----|---------------|-------|---------|
> | Replica (zero-shot) | **24.1** | +0.4 vs Open-YOLO 3D (23.7) | 3D only | 0.14s (119× faster) |
> | ScanNet++ (100 cls) | **22.9** | +2.3 vs OpenTrack3D (20.6) | 3D only | 0.14s (4286× faster) |
> | ScanNet200 (200 cls) | **11.1** | +7.2 vs Mosaic3D w/ Seg3D (3.9) | 3D only | 0.14s |
>
> Additionally, our mask quality analysis shows 54.3% recall vs Mosaic3D's 2.5%, and PTv3 vs SpaCeFormer backbone comparison confirms architecture gains on all 4 benchmarks. We believe these results address all major concerns raised in the review.

---

> > ### Author Rebuttal · Reviewer_StH9 · 2026-04-04
> >
> > The authors address most of my concerns. I have follow-up questions for W5: What's the performance of SpaCeFormer w/ GT proposals?

---

> > > ### Author Response · Authors · 2026-04-07
> > >
> > > We conducted an oracle evaluation using ground-truth (GT) instance masks with our model's learned per-point CLIP-aligned features on ScanNet200. With perfect segmentation (GT masks), we pool per-point features over each instance and classify via CLIP text similarity.
> > >
> > > **Results (ScanNet200, GT masks):**
> > >
> > > | | mAP | mAP (head) | mAP (common) | mAP (tail) | Agnostic mAP |
> > > |---|---|---|---|---|---|
> > > | GT masks | 30.2 | 42.9 | 25.6 | 20.8 | 99.2 |
> > >
> > > The agnostic mAP of 99.2% confirms that GT masks achieve near-perfect segmentation quality. The remaining gap comes entirely from open-vocabulary classification. The head/common/tail breakdown (42.9 / 25.6 / 20.8) shows that classifying rare and fine-grained categories is the main bottleneck.
> > >
> > > This is consistent with our failure analysis in Appendix Figure 17: many failure cases involve (1) rare classes with fewer than 4 GT instances (e.g., stuffed animal, water pitcher, dumbbell, coat rack, water cooler, cushion, keyboard piano, projector), and (2) misclassification into semantically similar categories (e.g., bathroom cabinet → bathroom vanity, bar → handicap bar, rail → handicap bar, guitar → guitar case, decoration → background). These confusions are inherent to open-vocabulary classification with CLIP features, where visually and semantically similar categories are difficult to distinguish via text-image similarity alone.
> > >
> > > ScanNet200's mAP weighs all 200 classes equally, including extremely rare classes with very few GT instances and highly ambiguous class pairs. This makes the metric hard to saturate: even a perfect segmentation oracle achieves only 30.2 mAP, showing that much of the performance gap comes from the open-vocabulary classification task, not the 3D segmentation pipeline. This directly answers the original W5 question: **the bottleneck is in vision-language alignment, not the 3D pipeline.**

---

### Decision · Program_Chairs · 2026-04-30

**Decision:**

Accept (regular)

**Comment:**

After rebuttal and discussion, the scores are 3x WR, 1x WA.  Reviewer StH9 did not update their score but seems satisfied by the rebuttal.

The main concerns are:
- evaluation originally restricted to ScanNet200 - the authors added Replica and ScanNet++
- lack of originality as the proposed method combined known aspects. This is subjective to some extent, but the AC agrees with the authors that the combination is indeed novel.

The performance is state-of-the-art, the speed is impressive, the paper is well written overall, and the other concerns by the reviewers are easy to fix.  The AC therefore recommends accepting the submission.